# LRRT*: A robotic arm path planning algorithm based on an improved Levy flight strategy with effective region sampling RRT*

Yu Gu[1], Hua Luo[2]*, Wenbin Gong[1], Yutao Jiang[1], Tangju Yuan[1], Longzhou Cao[1]*, Hongbing Li[2,3], Wei Zhang[2,3]

**1** Chongqing Key Laboratory of Geological Environmental Monitoring and Disaster Early Warning in the Three Gorges Reservoir Area, Chongqing Three Gorges University, Wanzhou, Chongqing, China, **2** Internet of Things and Intelligent Control Technology Chongqing Engineering Research Center, Chongqing Three Gorges University, Wanzhou, Chongqing, China, **3** Chongqing Municipal Key Laboratory of Intelligent Information Processing and Control, Chongqing Three Gorges University, Wanzhou, Chongqing, China

☯ These authors contributed equally to this work.
* 1915854548@qq.com (HL); 29209645@qq.com (LC)

**Data availability statement:** All data files are available from the Harvard Dataverse database (accession number(s) https://doi.org/10.7910/DVN/BUWWAA).

## Abstract

Aiming at the problems of blind sampling points and slow planning speed of path planning Rapidly-exploring Random Trees algorithm, an effective region sampling Levy Rapidly-exploring Random Trees algorithm (LRRT*) is proposed based on the improved Levy flight strategy. Divide the entire path planning process into two stages: quickly finding the initial path and optimizing the path. Goal oriented strategy is used to explore the path when finding the initial path quickly. The Levy flight strategy is used to regenerate nodes after obstacles are encountered to improve the quality of the expansion points. They can quickly plan a collision-free path. In the phase of optimizing the initial path using the effective region sampling method, each sampling is only sampled around the initial path. Meanwhile, node rejection strategy is introduced to reduce the number of collision detection and accelerate the convergence speed. In 2D and 3D environments, the LRRT* algorithm reduces the initial path planning time by 17.6% and 91.9% respectively compared to the RRT* algorithm, and shortens the average planning time by 12.3% and 65.5%, and the path smoothness is 3.4% and 79.4% shorter respectively. Applying the LRRT algorithm to a robotic arm allows for the planning of collision-free paths.

## 1 Introduction

With the continuous development of agriculture to modernization and intelligence, more and more robots are applied to the field of agriculture. Fruit and vegetable picking is an important agricultural activity, and traditional picking operations are mainly done manually, which is

**Funding:** The work was supported by National Key R&D Program of China (2021YFB3901400), Natural Science Foundation of Chongqing Municipality (2022NSCQ-MSX4084), Scientific and Technological Research Program of Chongqing Municipal Education Commission (KJZD-M202201204, KJZD-M202301203, KJQN202301215, KJQN202401226, 22SKGH333), Open Fund of Chongqing Key Laboratory of Geo-environment Monitoring and Disaster Early Warning of Three Gorges Reservoir Area (MP2020B0202), Science and Technology Innovation Smart Agriculture Project of Wanzhou District Science and Technology Bureau (2022-17), Cultivation project of Chongqing social science planning project (2019PY52).

also the most energy-consuming process. At present, the agricultural labor force is constantly losing, labor costs are rising, and some high-altitude fruits pose a great danger to picking. Fruit and vegetable picking is not only a labor-intensive and time-consuming work, but also a dangerous work [1]. Picking robotic arm is the key part of the picking robot, which is a big hot spot in the research of picking robot [2].

One of the key technologies of the picking robotic arm is the path planning. A process in which the robotic arm moves from the starting position to the target position without colliding with obstacles in between [3]. The short harvest cycle, fragile outer skin and random growth location of fruits bring great difficulty to the robotic arm path planning [4]. The convergence speed and stability of path planning determine whether the picking robotic arm can efficiently and accurately complete the picking task. Therefore, extensive research has been conducted by scholars both domestically and internationally on algorithms for robotic arm path planning.

Robotic arm path planning algorithms can be divided into genetic algorithms and particle swarm algorithms based on swarm optimization theory, artificial potential field algorithms based on optimization theory, A* algorithms and Dijkatra algorithms based on graph search algorithms, and Rapidly-exploring Random Trees (RRT) based on sampling [5]. The first three types of algorithms are usually applied in the path planning of mobile robots, but they are poorly suited for path planning in high-dimensional spaces such as robotic arms.

Swarm intelligence-based optimization algorithms exhibit notorious susceptibility to local optima entrapment in complex obstacle environments, compounded by excessive parameter sensitivity. For instance, in robotic manipulators with high degrees of freedom (DoFs), the dimensional explosion of solution space fundamentally undermines population-based methods like Genetic Algorithms (GA). These algorithms necessitate maintaining large-scale populations (typically hundreds to thousands of individuals), with each iteration requiring selection, crossover, and mutation operations—a process whose temporal complexity escalates exponentially with dimensionality, thereby failing to meet real-time constraints.

The Artificial Potential Field (APF) approach faces dual limitations: motion stagnation occurs when the resultant vector of attractive and repulsive fields nullifies at non-target configurations, while its hyper-sensitive parameter tuning requirements and poor generalizability hinder practical deployment. Furthermore, graph-search algorithms like A* and Dijkstra encounter dimensional catastrophe in high-dimensional configuration spaces. The discretized node count grows exponentially with dimensionality, rendering these methods computationally prohibitive for robotic applications requiring.

Research shows that the sampling-based algorithm, RRT algorithm, is more suitable for robotic arm path planning [6]. Sampling-based RRT algorithm has the advantages of simple structure, strong search ability, and can be applied to high-dimensional space. It is widely used in picking robotic arm path planning, but it has the disadvantage that the path is not optimal [7]. Karaman et al. proposed RRT* algorithm to find an asymptotically optimal path through the process of re-selecting the parent node and rewiring [8]. Although the RRT* algorithm is able to find an asymptotically optimal path, its convergence time is long.

The current RRT* algorithm still has the following problems: (1) The RRT* algorithm has too many nodes rounded off due to the failure of collision detection, so the effectiveness of the nodes is poor. (2) RRT* algorithm is capable of finding an asymptotically optimal path, yet it comes with a considerable time cost. RRT* samples mostly in the invalid region. Excessive redundant nodes result in an increase in the number of collision detections during pruning, leading to slow convergence speed.

Aiming at the problems of poor node effectiveness and slow convergence of RRT* algorithm, this paper proposes the LRRT* algorithm incorporating Levy flight strategy and effective region sampling. The main contributions of this work are as follows.

(1) During the swift initial stage of path finding, the target is sampled with a predefined probability, simultaneously undergoing random spatial sampling. This accelerates the expansion of nodes to the target point. And a Levy flight is performed on the nodes that fail collision detection to re-generate new nodes to improve the effectiveness of the nodes.

(2) In the stage of optimizing the initial path, the entire map is divided into 16 equal parts. According to the distribution of initial path points in each partitioned area, the 16 areas are divided into effective sampling areas and invalid sampling areas. Thereafter sampling is performed only in the effective region, so as to continuously optimize the initial path, reduce the generation of invalid nodes, and reduce the path cost.

(3) Node rejection strategy is introduced to remove nodes with higher path cost. This reduces the number of collision detection during pruning and optimizing the convergence speed.

This work is arranged as follows. Section 2 presents the research progress of the robotic arm path planning algorithm. Section 3 introduces the robotic arm simulation model, the robotic arm collision detection model, the RRT* algorithm and the proposed LRRT* algorithm. Section 4 provides the comparative experiments of the RRT* algorithm, the AS-RRT* algorithm, the MQ-RRT* algorithm and the LRRT* algorithm, and verifies the effectiveness of the LRRT* algorithm on the robotic arm. Section 5 is the summary.

## 2 Related works

### 2.1 Rapidly-exploring random trees algorithm (RRT)

Rapidly-exploring Random Trees (RRT) algorithm is a single query-based algorithm based on data structure, primarily used in the direction of path planning, virtual reality and other research. The rapidly-exploring Random Trees algorithm is constructed using a special incremental approach. So the algorithm has a promising application in the field of picking robotic arm path planning. RRT* algorithm is an excellent variant of RRT algorithm. Its main feature is to find the initial path and then continuously optimize the path until it reaches the set maximum cycle time or time [9]. However, the RRT* algorithm still suffers from issues such as poor node effectiveness and slow convergence speed. Many scholars have made some improvements for these problems.

Gammell et al. [10] addressed the issue of blind sampling in RRT* algorithm by proposing the Informed RRT* algorithm. By restricting the sampling area to an ellipsoid, it reduces ineffective sampling. However, this approach does not limit the size of the random tree and is prone to getting trapped in local minima. To improve the convergence speed, Gammell et al. [11] introduced the concept of dual trees and proposed Bi-RRT* algorithm. Random trees are generated at the start and end points respectively to improve search speed and continuously optimize the path, but its path planning success rate is relatively low. Liu Xueshen et al. [12] introduced a node rejection strategy and optimized the selection range of parent nodes to propose an improved RRT* algorithm, aiming to address the shortcomings of RRT* algorithm such as long execution time and high memory usage. However, this improved algorithm suffers from issues like high path cost and low planning success rate. Zhang Qin et al. [4] proposed CTB-RRT* algorithm for the issues of random sampling and slow expansion of

RRT* algorithm. It reduces the blindness of sampling by replacing random uniform sampling with Cauchy distribution sampling strategy. Luan Qinglei et al. [13] proposed the AS-RRT* algorithm. It alleviates computational pressure by introducing an adaptive goal-bias strategy and a node rejection strategy. It utilizes Sobol sequences to generate uniformly distributed sampling points and adopts a sliding sampling pool method for sampling, thereby improving the quality of the path.

In addition to improving the sampling strategy as well as reducing the tree nodes, the convergence time can be shortened by adding quality parent nodes. Xining Cui et al. [9] proposed an MQ-RRT* algorithm based on optimizing sampling points. The algorithm proposes a method to create new parent nodes, which improves the quality of the initial path and the rate of convergence to the optimal solution. Liao et al. [14] proposed an F-RRT* algorithm to solve the problem of difficult generation of nodes in close proximity to obstacles. The algorithm optimizes the tree by creating nodes closer to the obstacles, improving the quality of the initial solution and speeding up the rate of convergence.

The LRRT* algorithm proposed in this paper differs from the aforementioned algorithms in that it does not use elliptical regions to limit the sampling scope. Instead, it divides the area into blocks and samples within the effective regions. Additionally, the introduction of the Levy flight strategy enhances the efficiency of the nodes.

## 2.2 Swarm optimization algorithm

The principle of swarm optimization algorithms is to set multiple individuals in the space and form a population of these individuals to search and get the optimal solution [15]. Common swarm optimization algorithms are Particle Swarm Optimization Algorithm (PSO), Ant Colony Algorithm (ACA) and Genetic Algorithm (GA). Yuan Monn et al. proposed a multiple swarm particle swarm algorithm based on monocular vision. This algorithm combines the elite population with the subpopulation to form a multi-population particle swarm. The pre-selection and interaction mechanism is used to enable the algorithm to jump out of the local optimum. The optimal trajectory of the robotic arm is determined based on the target position [16]. Ding et al. [17] proposed an improved ant colony-sequential local search path planning strategy in order to reduce the sum of path length and joint rotation angle at the end of the robotic arm. Compared with the traditional ant colony, the path planning length is reduced by 3.26% and the sum of joint rotation angles is reduced by 2.21%. Aiming at the low efficiency of traditional robotic arm path planning algorithm, Huaijiang Wang et al. [18] proposed a mobile robotic arm sorting path optimization algorithm based on improved genetic algorithm. The algorithm uses improved rank evolutionary selection operator and optimal nearest neighbor crossover algorithm. The path is shortened by 45.99%, and the system running time is reduced by 25.80%, which improves the system efficiency.

## 2.3 Artificial potential field algorithm

Artificial potential field algorithm is a classical robot path planning algorithm. The algorithm regards the target and obstacles as objects that have gravitational and repulsive forces on the robot, respectively. The robot moves along the combined force of gravitational force and repulsive force [19]. Tiantian Miao et al. proposed a shortest obstacle avoidance path algorithm for robotic arms based on the gravitational factor of the artificial potential field for the problem of safe obstacle avoidance of robotic arms. The algorithm utilizes the gradient descent method to solve the shortest obstacle avoidance path and its optimal solution at the end of the robotic arm. It enables the end of the robotic arm to avoid obstacles along the shortest path while its motion trajectory can be tangent to the obstacles [20].

Yanjie Li et al. [21] proposed the PQ-RRT* algorithm by combining the artificial potential field algorithm and the RRT* algorithm. This algorithm can obtain new nodes that are closer to the expected ones, thereby reducing the time to expand to the target area. Caio Cristiano [22] combined adaptive artificial potential field algorithm with end-effector directional control technique, where the force generated by the adaptive artificial potential field guides the robot end-effector to the target for real-time robotic arm control system.

## 2.4 Graph search algorithm

Graph search algorithms rely on known environment maps and obstacle information to construct a feasible path, and the commonly used ones are Dijkasta algorithm and A* algorithm. Yabin Zhang et al. [23] used the Dijkasta algorithm for three-degree-of-freedom robotic arm path planning to derive the shortest path and verified the feasibility of the algorithm. Tang et al. [24] proposed an obstacle-avoidance path algorithm for a six-degree-of-freedom robotic arm based on the improved A* algorithm. The enhanced A* algorithm proposes a new node search strategy and local path optimization method, which significantly reduces the number of search nodes and improves the search efficiency. Zhang et al. [25] combine genetic algorithm and A* algorithm to achieve obstacle avoidance by using the A* algorithm on paths obtained by the genetic algorithm. The feasibility of the algorithm was verified through simulations and experiments.

# 3 Methods

## 3.1 Model

### 3.1.1 Robotic arm model.
UR5 robotic arm, manufactured by Universal Robots, serves as the robotic arm model employed in this paper, as shown in Fig 1.

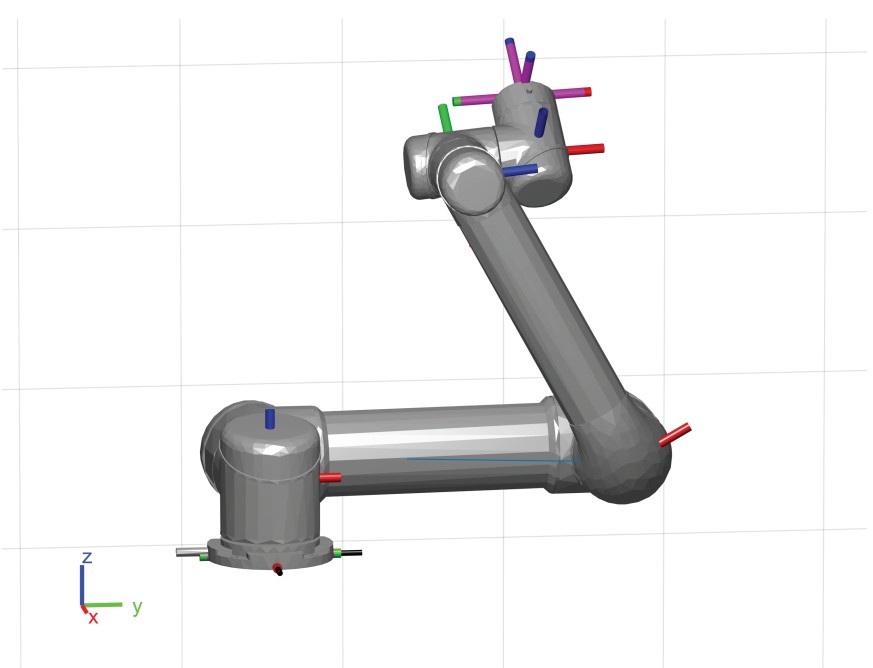

**Fig 1. UR5 robotic arm.**

The DH method is used to model the UR5 manipulator, and the parameters of each connecting rod and joint are shown in Table 1.

Where $i$ represents the link number. $\alpha$ is the twist angle between adjacent links. $a$ is the length of the link. $d$ denotes the linkage deflection, and $\theta$ is the rotation angle of the joint. The positive kinematics equations can be derived from Table 1 and the transformation matrix of Eq (1), as shown in Eq (2). The positive kinematics equation can be utilized to determine the position of the end-effector of the manipulator within the Cartesian space. $n$, $o$, $a$ denoting the attitude of the end-effector of the manipulator, $p$ denoting the position of the end-effector of the manipulator.

$$
{}_i^{i-1}T = \begin{bmatrix} \cos\theta_i & -\sin\theta_i & 0 & \alpha_{i-1} \\ \sin\theta_i\cos\alpha_{i-1} & \cos\theta_i\cos\alpha_{i-1} & -\sin\alpha_{i-1} & -d_i\sin\alpha_{i-1} \\ \sin\theta_i\sin\alpha_{i-1} & \cos\theta_i\sin\alpha_{i-1} & \cos\alpha_{i-1} & d_i\cos\alpha_{i-1} \\ 0 & 0 & 0 & 1 \end{bmatrix} \tag{1}
$$

$$
{}_6^0T = {}_1^0T\,{}_2^1T\,{}_3^2T\,{}_4^3T\,{}_5^4T\,{}_6^5T = \begin{bmatrix} n_x & o_x & a_x & p_x \\ n_y & o_y & a_y & p_y \\ n_z & o_z & a_z & p_z \\ 0 & 0 & 0 & 1 \end{bmatrix} \tag{2}
$$

**3.1.2 Obstacle collision detection model.** In actual working environments, the shapes of obstacles are complex, so it requires a significant amount of time to determine whether the robotic arm collides with the obstacles. Therefore, the robotic arm and obstacles are usually enveloped to make them regular shapes. The robotic arm is simplified as a cylinder and the obstacles are simplified as cuboids. Although this increases the size of the model, it simplifies the collision detection and reduces the possibility of collisions of the robotic arm, as shown in Fig 2.

The robotic arm linkage is regarded as a cylinder with a $r$ radius of so that the robotic arm can be further simplified to a line segment AB. The collision detection between the manipulator and irregular obstacles can be simplified to the collision detection between line segments and regular obstacles. The dotted line in Fig 2 represents the actual obstacle, which will be rectangular envelope and expansion processing, the expansion size of $r$. Obstacle envelope processing as shown in Eq (3), and the equation of the line segment AB is set to be $f(x, y, z)$. If the robotic arm collides with obstacles, the solution of $f(x, y, z) = 0$ has an intersection with Eq (3).

$$
\begin{cases} x_{min} + r \le x \le x_{max} + r \\ y_{min} + r \le y \le y_{max} + r \\ z_{min} + r \le z \le z_{max} + r \end{cases} \tag{3}
$$

**Table 1. DH parameters.**

| $i$ | $\alpha_{i-1}/rad$ | $a_{i-1}/mm$ | $d_i/mm$ | $\theta_i/rad$ |
|---|---|---|---|---|
| 1 | $\pi/2$ | 0 | 162.5 | 0 |
| 2 | 0 | −425 | 0 | 0 |
| 3 | 0 | −392.2 | 0 | 0 |
| 4 | $\pi/2$ | 0 | 133.3 | 0 |
| 5 | $-\pi/2$ | 0 | 99.7 | 0 |
| 6 | 0 | 0 | 99.6 | 0 |

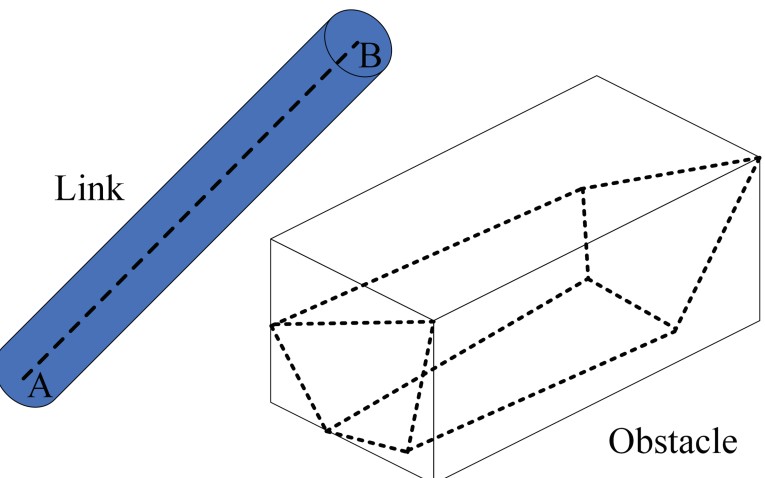

**Fig 2. Robotic arm linkage envelope and obstacle envelope.**

### 3.2 RRT*

RRT algorithm is a sampling-based path planning algorithm [26]. RRT* algorithm is an improvement upon RRT algorithm, incorporating the steps of reselecting parent nodes and rewiring. Its principle is as follows. First, create a tree T that contains the starting point (X_init) and mark the starting point as the root node. Randomly sample a point (X_rand) in the search space and find the nearest node (X_near) in the tree. Then, try to extend a branch with a fixed step length from the nearest node to obtain a new node (X_new). If the new node is within the feasible region of the search space, add it to the tree. Repeat the above process until the random tree expands to the target point (X_goal), and all nodes on the random tree are saved in the node table P, as shown in Fig 3, with black shapes indicating obstacles.

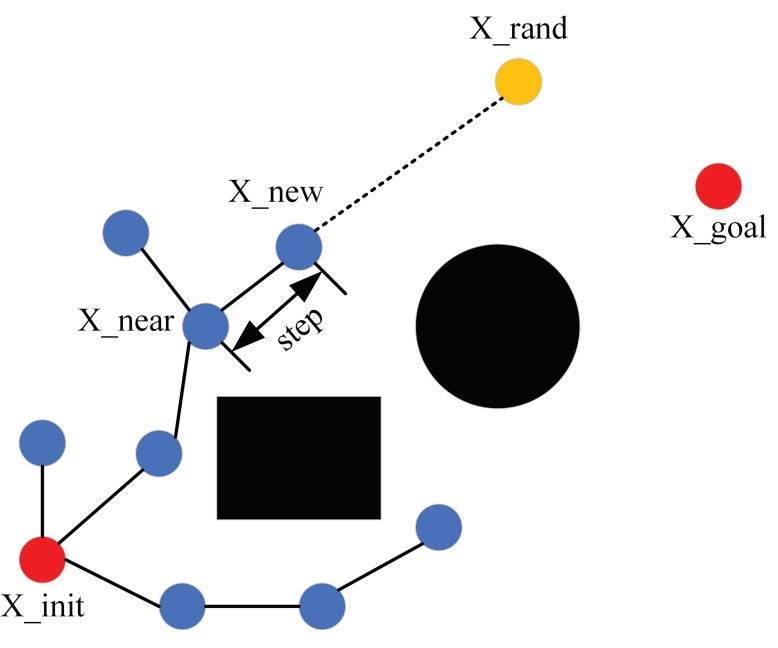

**Fig 3. RRT*.**

RRT* algorithm incorporates a process of revisiting parent node selection and rewiring during the expansion of new nodes, ensuring that the paths it generates are asymptotically optimal in nature. Reselecting the parent node is shown in Fig 4. For the new node extended by the random tree each time, the node with the minimum distance within a fixed range is taken as the parent node of X_new. Node 5 can serve as the new parent node of Node 9, and the path cost from the new node to the root node will be reduced from 14 to 11. The process of rewiring is shown in Fig 5. After reselecting the parent node, then calculate the distance from all the nodes in its given range to X_new. If any of the nodes has a smaller distance to X_new than to its original parent node, then update X_new to be the parent node of this node. Node 9 can be the new parent node of Node 6, and in this way, the path cost of Node 6 will be reduced from 15 to 12.

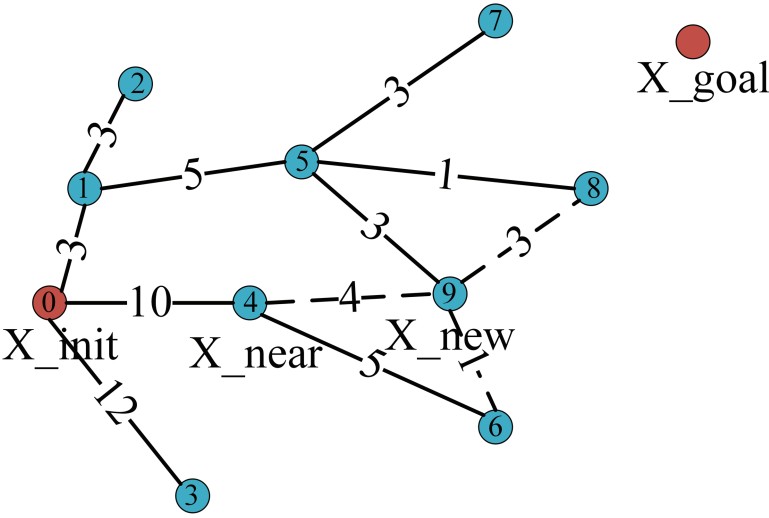

**Fig 4. Reselect parent node**.

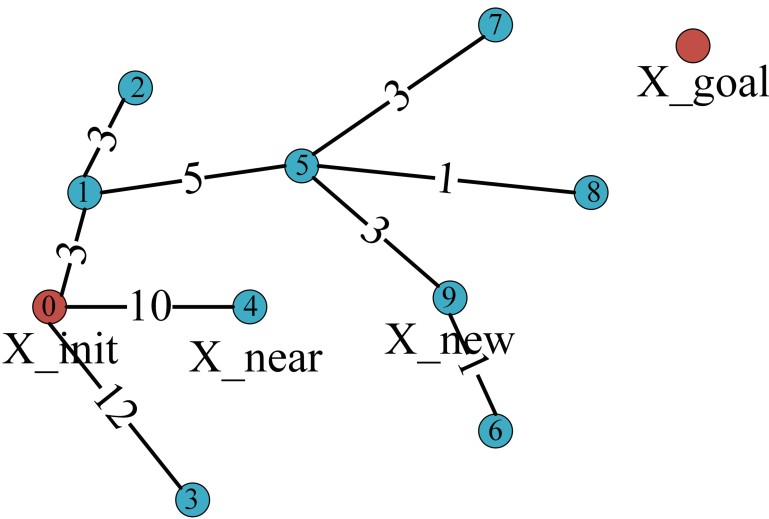

**Fig 5. Rewire**.

### 3.3 Improved RRT*

RRT* algorithm employs random sampling throughout the entire space for its entire process, resulting in a rather haphazard and inefficient approach for both path generation and optimization. In this work, RRT* algorithm is divided into two stages: fast planning initial path and optimizing path. During the stage of quickly finding the initial path, we add the goal-oriented strategy and Levy flight strategy. In the initial path optimization stage, we add the effective region sampling, and use the greedy idea to reduce the path points after getting the final path. Finally, a Bezier curve is added to smooth the curve.

**3.3.1 Goal oriented strategy.** RRT* algorithm performs random sampling in the entire space. When there are fewer obstacles, it will slow down the convergence speed of the algorithm. Therefore, a goal-oriented strategy is introduced, as shown in Eq (4) [27]. During sampling, the target point is considered as a sampling point with a fixed probabilit [9]. It guides the path to expand toward X_goal and accelerates the formation of the initial path.

$$X\_rand = \begin{cases} X\_goal & , r < p \\ rand & , r > p \end{cases} \tag{4}$$

where $X\_rand$ denotes the random point, $X\_goal$ denotes the target point, $p$ is the set target bias probability, usually set to 0.5. $r$ represents a random number and $r \subset [0, 1]$.

**3.3.2 Levy flight strategy.** RRT* algorithm discards a new point when it is generated that fails the collision detection, which will make it difficult for RRT* algorithm to pass obstacles quickly and will fall into a local minimum. Simply performing a Levy flight on a node that fails the collision detection will have a high probability of bypassing the obstacle and jumping out of the local minimum.

The Levy flight is a specialized random walk model used to characterize movement patterns with long-tailed distributions. In Levy flights, individuals perform stochastic movements in space, where both step lengths and directions are governed by the Levy distribution—a probability distribution exhibiting heavy-tailed characteristics. Its probability density function follows a power-law relationship, indicating that Levy-distributed motion exhibits significantly higher frequencies of large-step events (i.e., long-distance displacements) compared to Gaussian or other conventional distributions.

In addressing path planning challenges under static and dynamic environments, He Jianchen et al. [28] integrated the Levy flight strategy with the Dung Beetle Optimizer (DBO) and Dynamic Window Approach (DWA). Specifically, they modified the position update formulas for breeding and foraging dung beetles in the DBO algorithm using Levy flight dynamics, thereby enhancing the algorithm's exploration capability and adaptability. Correspondingly, Niu Yanbiao et al. [29] tackled the issue of poor dynamic obstacle avoidance in UAV path planning by proposing an enhanced Sand Cat Swarm Optimization (SCSO) algorithm. This improved framework incorporates an adaptive social neighborhood search mechanism and Levy flight strategies, which collectively elevate solution quality through refined exploration-exploitation balance.

Levy flight strategy is a combination of short-distance bouncing and occasional longer-distance walking. Short-distance bouncing ensures that the re-spawned new node is near $X\_rand$, reducing the probability of collision with obstacles. Occasional longer distance walking ensures that the regenerated new node jumps out of the region where the obstacle is

located. The longer walks when Levy is flying are generally manifested in 90° turns, as shown in Fig 6. 90° turns can better bypass the obstacles.

To calculate the search path for Levy flight, the formula proposed by Mantegna for modeling the Levy flight path is usually used, as shown in Eq (5) [30].

$$Levy(\beta) = \frac{\mu}{|v|^{\frac{1}{\beta}}} \tag{5}$$

where $Levy(\beta)$ is the flight path, and $\beta$ takes the value in the range of [0,2], and is generally taken as $\beta =1.5$ [31–33]. $\mu$ and $v$ are random numbers that follow the normal distribution of Eq (6). The standard deviation of the normal distribution corresponding to Eq (6) satisfies Eq (7).

$$\begin{cases} \mu \sim N(0,\sigma_\mu^2) \\ v \sim N(0,\sigma_v^2) \end{cases} \tag{6}$$

$$\begin{cases} \sigma_\mu = \left[ \frac{\Gamma \cdot (1+\beta) \cdot \sin(\Pi\beta/2)}{\Gamma[(1+\beta)/2] \cdot \beta \cdot 2^{(\beta-1)/2}} \right]^{\frac{1}{\beta}} \\ \sigma_v = 1 \end{cases} \tag{7}$$

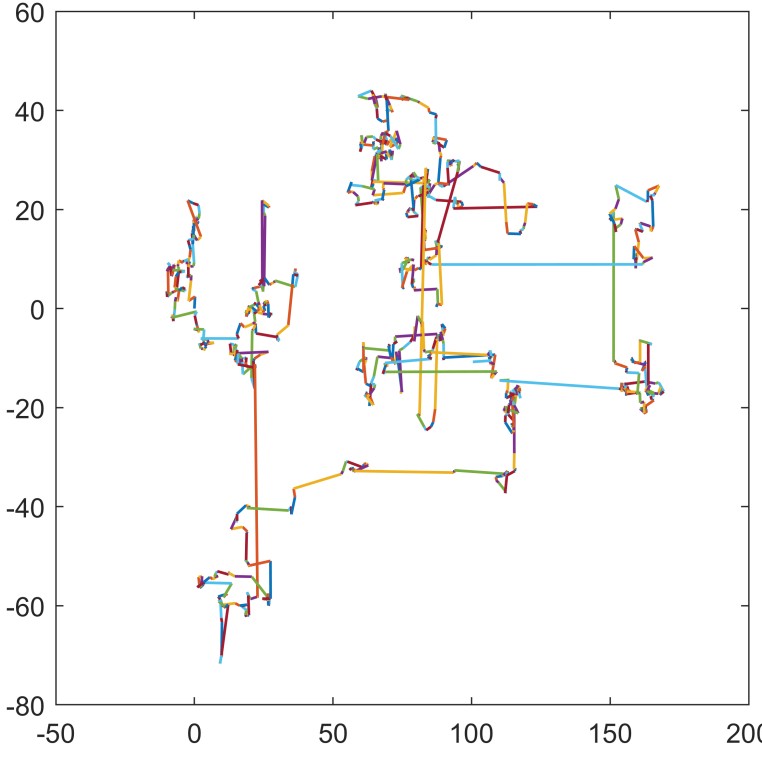

**Fig 6. Levy flight strategy.**

Levy flight strategy regenerates new nodes for nodes that fail collision detection according to Eq (8).

$$X\_nnew = X\_new + 0.1 \times step \otimes Levy(\beta) \tag{8}$$

where $X\_new$ denotes the node that fails the collision detection, $X\_nnew$ denotes the new node after Levy's flight, and $\otimes$ denotes the point-to-point multiplication.

**3.3.3 Effective region sampling strategy.** The whole process of RRT* algorithm involves random sampling in the given area, which produces a lot of invalid nodes. The effective sampling area in the actual space is very small, after generating the initial path there is no need to randomly sample in the whole space, only around the initial path. Based on the dimensionality and the effectiveness of segmentation, maps can be divided into $3^d$ blocks, where $d$ represents the dimensionality. As shown in Fig 7, a two-dimensional map is evenly divided into 9 blocks, with each block classified as either a valid or invalid sampling area depending on the presence of path points within the respective block. Similarly, in three dimensions, it can be divided into 27 blocks. In Fig 7, the green part indicates the initial path and the path points, and areas 1, 2, 5, 6 and 9 are effective sampling areas, while the rest are invalid sampling areas. Sampling only in the effective sampling region after forming the initial path can effectively reduce the generation of invalid nodes, and also accelerate the path optimization, which greatly shortens the path cost.

**3.3.4 Node rejection strategy.** Sampling is done in a certain range around the path, expanding only the tree towards the effective region. Each addition of a new node to the node

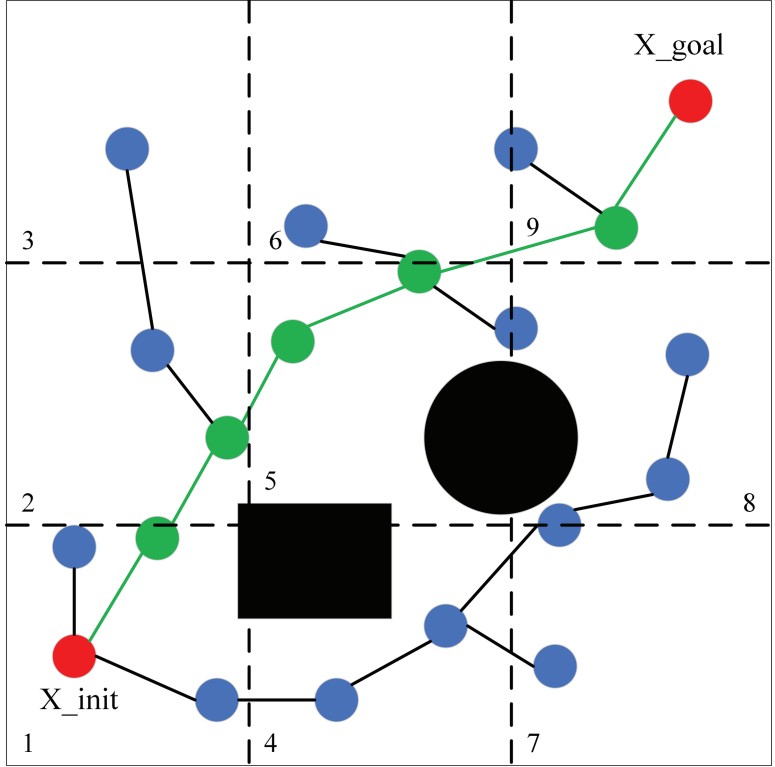

**Fig 7. Effective region sampling strategy.**

table necessitates the reselection of a parent node and rewiring. In the early stage the tree has fewer nodes and it takes less time to perform these two processes. However, in the later stage, an excessive number of nodes will cause these two processes to consume a lot of time, which is also an important reason for the slow convergence of the algorithm. So a node rejection strategy is introduced. As shown in Eq (9), if the cost of the path from X_new to X_init, plus the cost of the direct connection from X_new to X_goal, is smaller than the cost of the optimal path, then the new node is added to the tree. Otherwise, it is removed. In Fig 8, the cost of the path where X_new is located is 1+2+2+3=8, while the cost of the optimal path is 1+2+2+1+1=7, so the X_new node is deleted.

$$Dis(X\_new, X\_goal) + Dis(X\_new, X\_init) < 1.1 \times Dis\_min \tag{9}$$

In the equation, $Dis()$ represents the distance calculation function. $Dis\_min$ represents the cost of the current optimal path. The notation $1.1 \times Dis\_min$ indicates the goal of retaining potential good nodes as much as possible.

**3.3.5 Path node processing.** At the end of the path finding, a collision free path can be found and there will still be some redundant nodes on the path. As shown in Fig 9, nodes 2, 3 and 7 etc. are redundant nodes. The existence of these path points will lead to unstable operation of the robotic arm, adding greedy ideas. As long as there is no collision between two path points, delete the redundant path points in the middle, as shown in Fig 10. 1 and 4 nodes can be directly connected to each other with no collision, then delete 2 and 3 nodes, and similarly can be directly connected to 4 and 7, 7 and 9 nodes. The initial path has 10 nodes, after node processing, there are only 4 nodes in the path. It decreases the number of path points and shortens the path length.

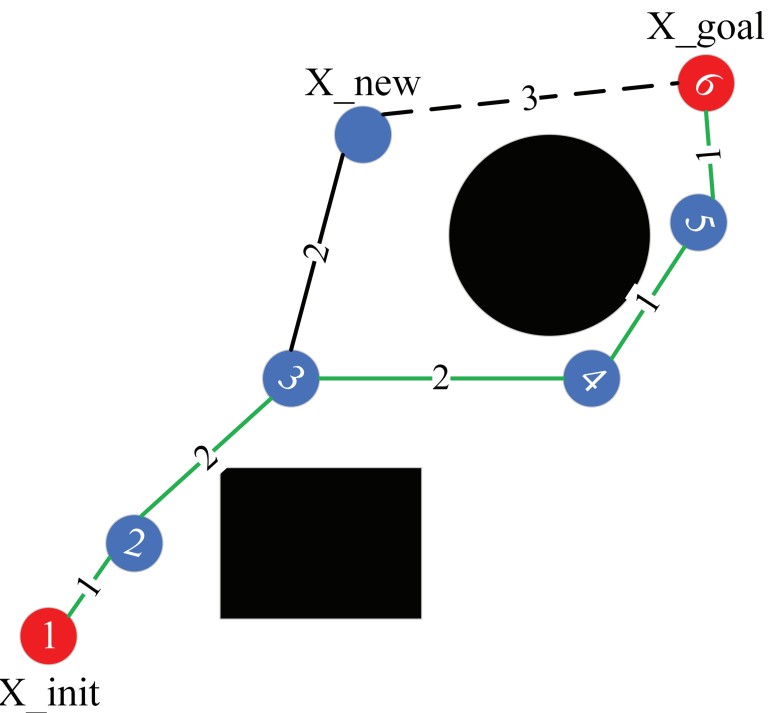

**Fig 8. Node rejection strategy**.

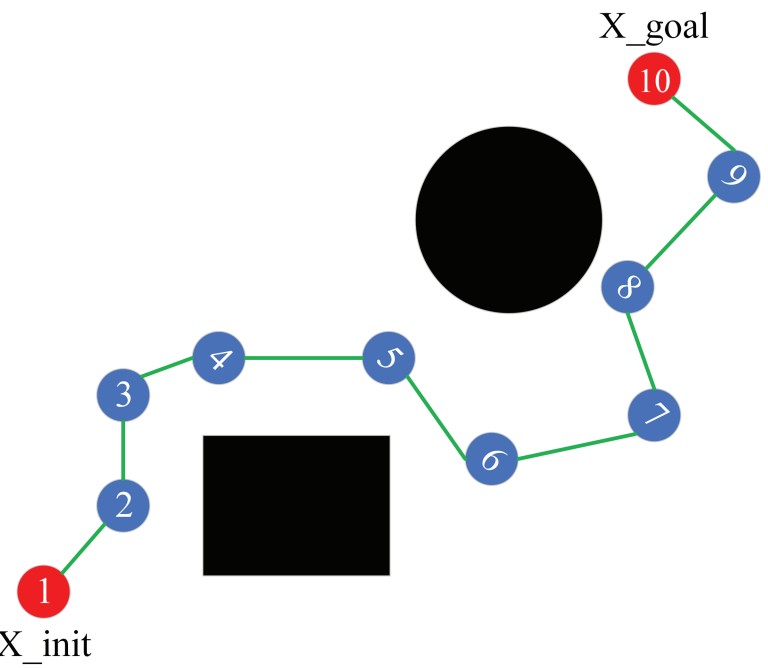

**Fig 9. Pre-greedy path.**

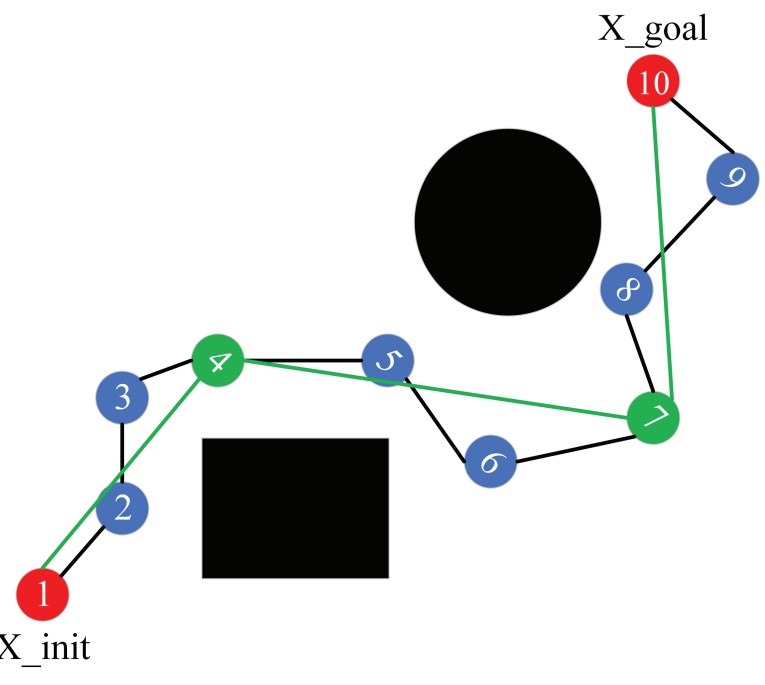

**Fig 10. Greedy back path.**

**3.3.6 Path smoothing.** After path optimization, the redundant nodes on the path have been eliminated. But the path is zigzag. A cubic B-spline curve can be used to smooth the path points, reducing the possibility of damaging the robotic arm due to jitters and other

issues during its operation. N times B-spline curve equation is shown in Eq (10) [34].

$$P_n(t) = \sum_{i=0}^{N} P_i G_{i,n}(t) \tag{10}$$

where $G_{i,n}(t)$ denotes the n-times B-spline basis function as shown in Eq (11) and Eq (12). $P_i$ denotes the control vertices, and the constituent polygons are the characteristic polygons of the B-spline curve.

$$G_{i,0}(t) = \begin{cases} 1 & t_i \le t \le t_{i+1} \\ 0 & otherwise \end{cases} \tag{11}$$

$$G_{i,n}(t) = \frac{t - t_i}{t_{i+n} - t_i} G_{i,n-1}(t) + \frac{t_{i+n+1} - t}{t_{i+n+1} - t_{i+1}} G_{i+1,n-1}(t) \tag{12}$$

In order to prevent the denominator from being 0, it is agreed that 0/0 = 0. The basis function of the 3 times B-spline curve is shown in Eq (13).

$$\begin{cases} g_0 = \frac{1}{6}\left(-t^3 + 3t^2 - 3t + 1\right) \\ g_1 = \frac{1}{6}\left(3t^2 - 6t + 4\right) \\ g_2 = \frac{1}{6}\left(-3t^3 + 6t^2 + 3t + 1\right) \\ g_3 = \frac{1}{6}t^3 \end{cases} \tag{13}$$

where $t$ denotes the path point and $g_0$ $g_3$ denotes the basis function.

The path after 3 times of B spline curve fitting process is shown in Fig 11. The black polyline is the original path, and the smoothed path is the green arc-shaped line, which can be seen that the path is smooth and can ensure the vibration-free operation of the robotic arm.

### 3.4 General flow of LRRT* algorithm

Step 1: Initialize the environment. Set parameters such as step, start position (X_init), goal position (X_goal), goal-directed probability, maximum number of iterations (max_iterations), pruning radius (RadiusForNeib), goal direct connection threshold (goal_region), and so on.

Step 2: Introduce a goal-directed strategy. Selects a random point (X_rand), finds the nearest node (X_near) the random point in the tree (T), and expands the new node (X_new) with a fixed step size.

Step 3: New node collision detection. If no collision is added to the tree, conversely a Levy flight is performed to regenerate X_new. Collision detection is performed again, the detection passes to be added to the tree, conversely discarded to return to step 2.

Step 4: Re-select parent node for X_new. Draw a circle with the X_new as the center and a step length as the radius, and determining whether any of these nodes can serve as the parent node for X_new.

Step 5: Rewire X_new. Judge whether the points within the ring can be used as child nodes of X_new.

Step 6: Determine whether the distance between X_new and X_goal is less than the target direct connection threshold. If it is less than the threshold then collision detection

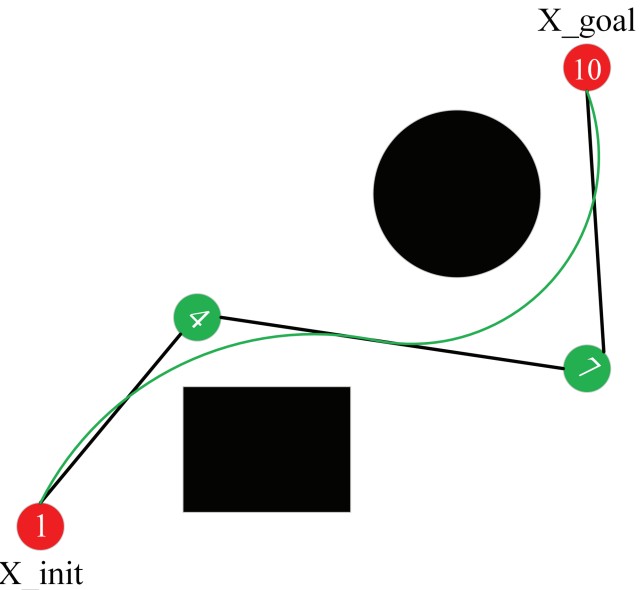

**Fig 11. Path smoothing**.

is performed. If the detection passes then direct connection to the target node is made. The initial path finding is completed. Instead return to step 2.

Step 7: After finding the initial path, the space is divided into valid and invalid regions based on the region where the path points are located. Thereafter sampling is done only in the valid region but without adding the goal oriented strategy and Levy flight strategy. The node rejection strategy is also added to remove the nodes whose path cost is too large.

Step 8: Check if the limit of maximum iterations, max_iterations, has been attained. After reaching max_iterations, the greedy idea is introduced to optimize the path points. The greedy path is optimized by B-spline curve smoothing for 3 times, and finally a smooth and stable curve is generated. The path planning is finished.

The pseudo-code of the LRRT* algorithm is shown in Algorithm 1. Among them, "sample()" is the random sampling function. "division()" is the partitioning function. "findnearestnode()" is the function for finding the nearest node. "extend()" is the function for expanding new nodes. "CollisionFree()" is the collision detection function. "ChooseParent()" is the function for reselecting parent nodes. "Rewire()" is the pruning function.

## 3.5 Algorithm analysis

Karaman et al. [8] defined the optimal path planning problem. Given a triad $(X\_init, X\_obs, X\_goal)$, $X\_init$ denotes the initial state, $X\_obs$ denotes the obstacle space, and $X\_goal$ denotes the goal state. Define $X\_free = X - X\_obs$ to denote the free space, where $X$ denotes the state space of the planned path, $\sigma : [0, 1] \to X\_free$ denotes a collision-free path, and $\sum$ denotes the set of all feasible paths.

**Algorithm 1. LRRT* algorithm.**

**Require:** Step, *X_init* (Start position), *X_goal* (Goal position), *p_goal* (Goal-directed probability = 0.5), *max_iterations*, Radius-ForNeib (Pruning radius), *goal_region* (Goal direct connection threshold)

**Ensure:** T (Tree representing the explored space)

1: $T \leftarrow \{X\_init\}$
2: **for** *index* = 1 to *max_iterations* **do**
3:     **if** findPath **then**
4:         $X\_rand \leftarrow \text{sample}(rand)$
5:     **else**
6:         $\text{division}(numDivisions, mapsize, path1)$
7:         $X\_rand \leftarrow \text{sample}(rand)$
8:     **end if**
9:     $X\_near \leftarrow \text{findnearestnode}(X\_rand, T)$
10:    $X\_new \leftarrow \text{extend}(X\_rand, X\_near, Step)$
11:    **if** CollisionFree($X\_near, X\_new$) **then**
12:        $ChooseParentnode \leftarrow \text{ChooseParent}(T, X\_new, RadiusForNeib)$
13:        $Rewirenode \leftarrow \text{Rewire}(T, X\_new, RadiusForNeib)$
14:    **else**
15:        $X\_new \leftarrow \text{Levy}(X\_new, X\_near)$
16:        **if** CollisionFree($X\_near, X\_new$) **then**
17:            $ChooseParentnode \leftarrow \text{ChooseParent}(T, X\_new, RadiusForNeib)$
18:            $Rewirenode \leftarrow \text{Rewire}(T, X\_new, RadiusForNeib)$
19:        **end if**
20:    **end if**
21:    $T \leftarrow T \cup \{X\_new\}$
22:    **if** disToGoal($X\_new, X\_goal$) **then**
23:        $T \leftarrow T \cup \{X\_goal\}$
24:        findPath $\leftarrow 1$
25:    **end if**
26: **end for**
27: **return** T

The optimal path planning problem can be defined as finding a path that connects the initial state and the goal state and minimizes the path cost $\sigma$, as shown in Eq (14) [35].

$$\sigma^* = arg \min_{\sigma \in \sum} \{c(\sigma) | \sigma(0) = X\_init, \sigma(1) = X\_goal, \forall t \in [0,1], \sigma(s) \in X\_free\} \tag{14}$$

Karaman et al. [8] have proved the probabilistic completeness of RRT* algorithm. The proposed LRRT* algorithm, like RRT* algorithm, traverses every possibility when the number of iterations approaches infinity to approach 1 in probability. Karaman et al. also proved the asymptotic optimality of RRT* algorithm. LRRT* algorithm is an improvement of RRT* algorithm, and thus inherits the asymptotic optimality as well. In LRRT* algorithm, a goal-oriented strategy is added, which can guide the path node to expand to the X_goal faster. A node rejection strategy is added, which reduces the time loss due to collision detection and accelerates the convergence speed, i.e.. It is proved that the LRRT* algorithm has fast convergence.

For the time complexity of LRRT* algorithm, LRRT* algorithm adds the goal-oriented strategy, Levy flight strategy, dynamic region sampling strategy and node rejection strategy to RRT* algorithm. The goal-oriented strategy and Levy flight strategy do not increase the number of sampling points, but improve the quality of sampling points, and therefore do not increase the time. Dynamic region sampling strategy and node rejection strategy did not increase the number of sampling points. They only sample in the effective space, and rejecte the nodes with higher path cost, which reduced the number of collision detection. Therefore did not increase the time complexity either. It can be seen from the above analysis that the computational complexities of the LRRT* algorithm and the RRT* algorithm are asymptotic, both being $O(kn)$. $k$ is usually related to factors such as the dimension and the size of the target area.

## 4 Algorithm simulation and results

To demonstrate the efficacy and advantages of LRRT* algorithm, a simulation is conducted, wherein its performance is evaluated against RRT* algorithm, MQ-RRT* algorithm(2024), and AS-RRT* algorithm(2024). MQ-RRT* algorithm improves the sampling mechanism [9], while AS-RRT* algorithm incorporates a node rejection strategy [13]. However, the sampling mechanism and node rejection strategy in LRRT* algorithm differ from those of MQ-RRT* and AS-RRT*. Therefore, they are selected for comparative experiments. Two cases of simple obstacles and complex obstacles are set up in a two-dimensional environment. A complex obstacle case is set up in a three-dimensional environment. The time to find the initial path, the path finding time and the number of path nodes of the compared algorithms are analyzed. Finally the ablation experiments are done in the 2D environment.

The smoothness of a path can be quantified by measuring the angular variation between consecutive path points. To systematically evaluate this property, we propose a path smoothness metric $S(x)$, defined as Eq (15).

$$S(x) = \sum_{i=2}^{n-1} |\theta_i - \theta_{i-1}| \tag{15}$$

Where $\theta_i$ denotes the angle between the $(i–1)$-th and $i$-th path segments. A higher $S(x)$ value indicates increased path tortuosity, while lower values correspond to smoother trajectories.

### 4.1 2D environment

Two different maps of simple obstacles and complex obstacles were set up in the 2D environment for simulation comparison, and the specific maps are shown in Fig 12. The sizes of the different maps are all 1000×1000. (50, 50) is the start point. (900, 900) is the target point. The step and the max_iterations are set to 100 and 1000 respectively, in which the black area is the obstacle and the white area is the free space. LRRT* algorithm, RRT* algorithm, MQ-RRT* algorithm and AS-RRT* algorithm are subjected to 50 experiments respectively and take the average value. The feasibility of LRRT* algorithm is verified by comparing the average path cost, average planning time and number of path nodes.

**4.1.1 Simple obstacle environments.** The simple obstacle environment is set with only one rectangular obstacle, which is used to verify the algorithm's ability to bypass the obstacle. From Fig 13, it can be concluded that RRT* algorithm, MQ-RRT* algorithm, AS-RRT* algorithm, and LRRT* algorithm are all able to find a feasible path. RRT* algorithm, MQ-RRT* algorithm, and AS-RRT* algorithm tend to randomly pick up points throughout the map,

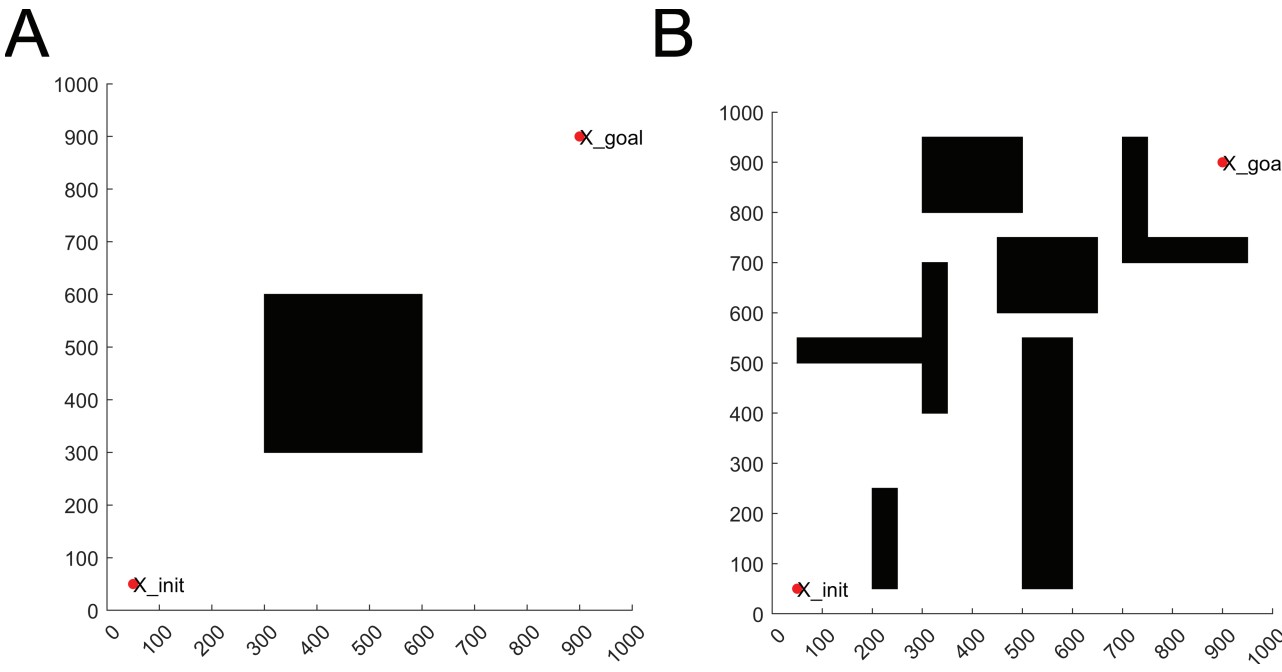

**Fig 12. 2D environment:** (**A**) simple obstacle environments. (**B**) Complex obstacle environments.

which is not conducive to optimizing the path. LRRT* algorithm, on the other hand, samples points around the feasible path, which is more conducive to optimizing the path. The paths of the four algorithms are generally not very different, but it is obvious that LRRT* algorithm has fewer branches and converges faster. Although AS-RRT* algorithm also has fewer branches, most of its sampling points are in the invalid sampling region, which is not conducive to optimizing the path. The red dots indicate the path points, and it can be concluded that LRRT* has significantly fewer path points than RRT*.

Fig 14 represents the relationship among path cost, number of iterations and time in simple obstacle environment. The folded line in Fig 14A represents the relationship between the path cost and the number of iterations. The bar graph represents the relationship between time and the number of iterations. In a simple obstacle environment, the proposed LRRT* algorithm has the minimum average path planning time under the condition that the path cost is similar to that of RRT* algorithm, MQ-RRT* algorithm, and AS-RRT* algorithm. A better path can be found more quickly. Fig 14B represents the path cost versus time. From Fig 14B, the time required for a path to reach a certain cost is less for LRRT* algorithm than RRT* algorithm and AS-RRT* algorithm. The higher the path cost requirement, the more obvious the contrast is. It indicates that LRRT* is more capable of finding the optimal path.

Different data pairs of the algorithm under 2500 iterations are shown in Table 2. The initial path planning time box-and-line plot and average planning time box-and-line plot are shown in Figure 17. In Fig 15A, LRRT* algorithm has the lowest box height, which proves that LRRT* algorithm has higher stability in initial path planning time than RRT* algorithm, MQ-RRT* algorithm, and AS-RRT* algorithm. It also has the lowest average value. In Fig 15B, LRRT* algorithm has the second highest box height after RRT* algorithm. It proves that it also has better stability in average planning time and its average value is the smallest. In summary, the comprehensive performance of LRRT* algorithm is better than RRT* algorithm,

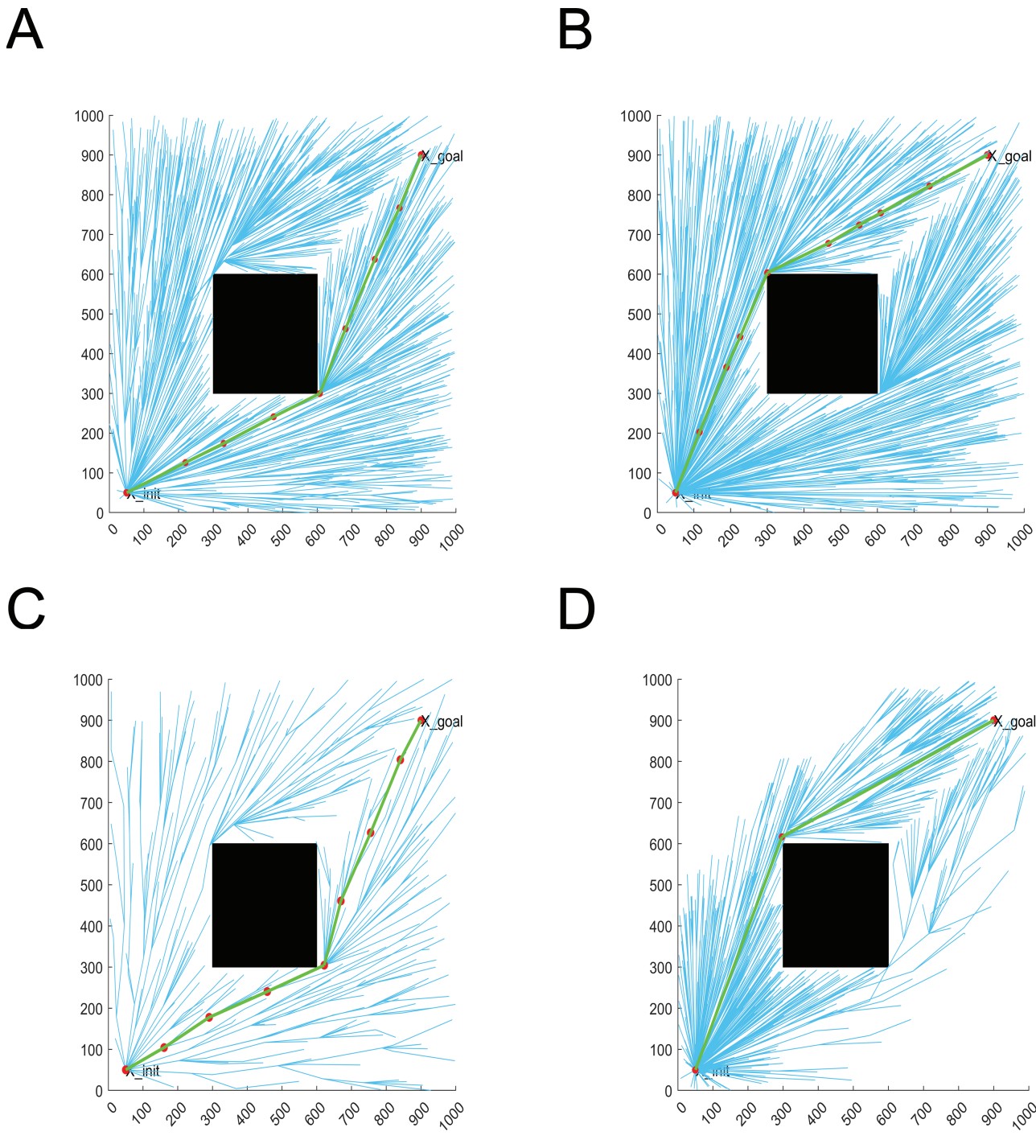

**Fig 13. Planning effect:** (**A**) RRT*. (**B**) MQ-RRT*. (**C**) AS-RRT*. (**D**) LRRT*

MQ-RRT* algorithm and AS-RRT* algorithm. From Table 2, LRRT* algorithm improves the initial path planning time by 61.6%, the average path planning time improves by 48.2%, the number of path nodes reduces by 66.3%, and the path smoothness is reduced by 24.2% than RRT* algorithm.

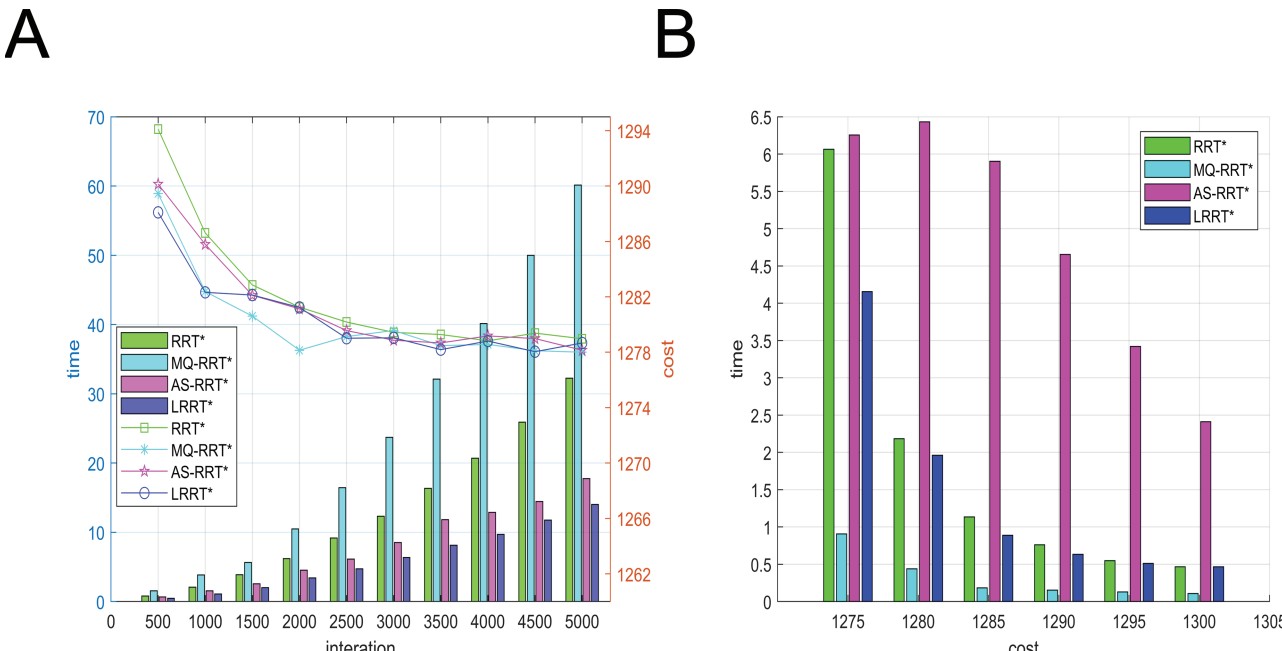

**Fig 14. Comparison of simulation results for simple obstacle environments:** (**A**) Path cost-time-number of iterations. (**B**) Path cost-time.

**Table 2. Simulation data of simple obstacle environment.**

| Algorithm | Initial path planning time/s | Average path planning time/s | Number of nodes | Smoothness |
|---|---|---|---|---|
| RRT* | 0.086 | 9.179 | 10.1 | 53.58 |
| MQ-RRT* | 0.047 | 10.971 | 9.0 | 45.43 |
| AS-RRT* | 0.104 | 6.266 | 10.0 | 47.04 |
| LRRT* | 0.033 | 4.760 | 3.4 | 40.63 |
| Improvement/% | 61.6 | 48.2 | 66.3 | 24.2 |

**4.1.2 Complex obstacle environments.** In a complex obstacle environment, there are a large number of obstacles with different sizes, shapes, and distances between them. This is used to verify the algorithm's adaptability in passing through irregular areas and in environments with a large number of obstacles. Path planning are shown in Fig 16. From Fig 16, it can be concluded that RRT* algorithm, MQ-RRT* algorithm and AS-RRT* algorithm take points in space more randomly, which is better for global exploration but poor for local exploitation. LRRT* algorithm gathers points around the paths in space, and also has points in other regions. It takes into account both global exploration and local exploitation. Moreover, compared with RRT* algorithm, MQ-RRT* algorithm and AS-RRT* algorithm, LRRT* algorithm has fewer sampling points. They are concentrated in the effective sampling region, reducing the number of collision detection, i.e., accelerating the convergence speed.

As shown in Fig 17, the relationship between the path cost and the number of iterations, planning time and the path cost of different algorithms in a complex obstacle environment are represented, respectively. From Fig 17A, LRRT* algorithm has faster planning time and smaller path cost. The planning time is less than RRT* algorithm and MQ-RRT* algorithm and slightly inferior to AS-RRT* algorithm. LRRT* algorithm outperforms RRT* algorithm and AS-RRT* algorithm and is slightly inferior to the MQ-RRT* algorithm in terms of path

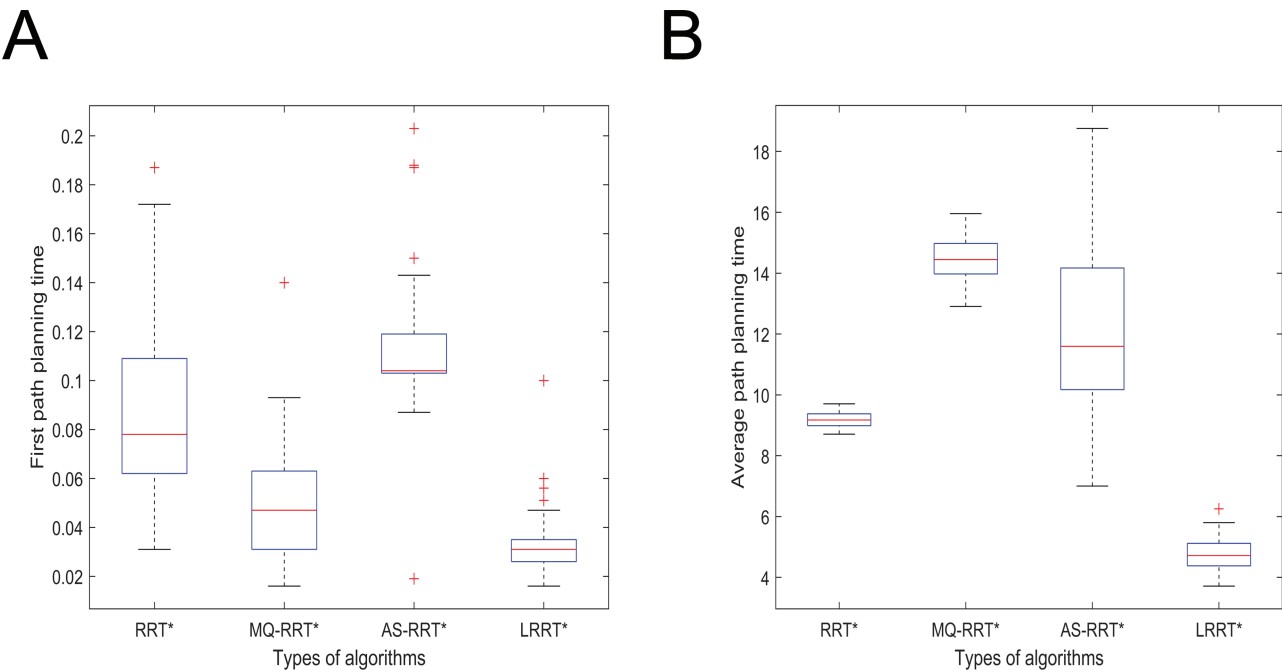

**Fig 15. Planning time for the 2500 iteration:** (**A**) Initial path planning time. (**B**) Average path planning time.

cost. From Fig 17B, LRRT* algorithm requires only a shorter time to plan a path with less path cost, and the higher the path cost requirement, the more obvious the contrast is.

The planning time boxplots of the simulation results of different algorithms under 3000 iterations are shown in Fig 18 and the data pairs are shown in Table 3. In Fig 18A, LRRT* algorithm has the lowest box height. It proves that LRRT* algorithm has higher stability in initial path planning time than RRT* algorithm, MQ-RRT* algorithm, and AS-RRT* algorithm. It also has the lowest average value. In Fig 18B, the box height of LRRT* algorithm is not much different from that of RRT* algorithm and AS-RRT* algorithm. It proves that it also has better stability in the average planning time and its average value is second only to AS-RRT* algorithm. In summary, the comprehensive performance of LRRT* algorithm is better than RRT* algorithm, MQ-RRT* algorithm and AS-RRT* algorithm. From Table 3, LRRT* algorithm improves the initial path planning time by 43.3%, the average path planning time improves by 30.7%, the number of path nodes reduces by 38%, and the path smoothness is reduced by 3.4% than RRT* algorithm.

## 4.2 3D environment

A moderate number of obstacles are set up in a three-dimensional environment. The obstacles are set up as spheres for ease of computation, with the aim of verifying the high-dimensional adaptability of the proposed algorithms. The path planned by the path planning is shown in Fig 19. From Fig 19, RRT* algorithm, MQ-RRT* algorithm and AS-RRT* algorithm take points in the space more randomly, spreading over the whole space. LRRT* algorithm takes points in the space clustered around the initial path. While exploring globally, it better conducts local exploitation. Moreover, compared with RRT* algorithm, MQ-RRT* algorithm and AS-RRT* algorithm, LRRT* algorithm are concentrated in the effective sampling region. It proves that it is also well adapted in 3-dimensional space.

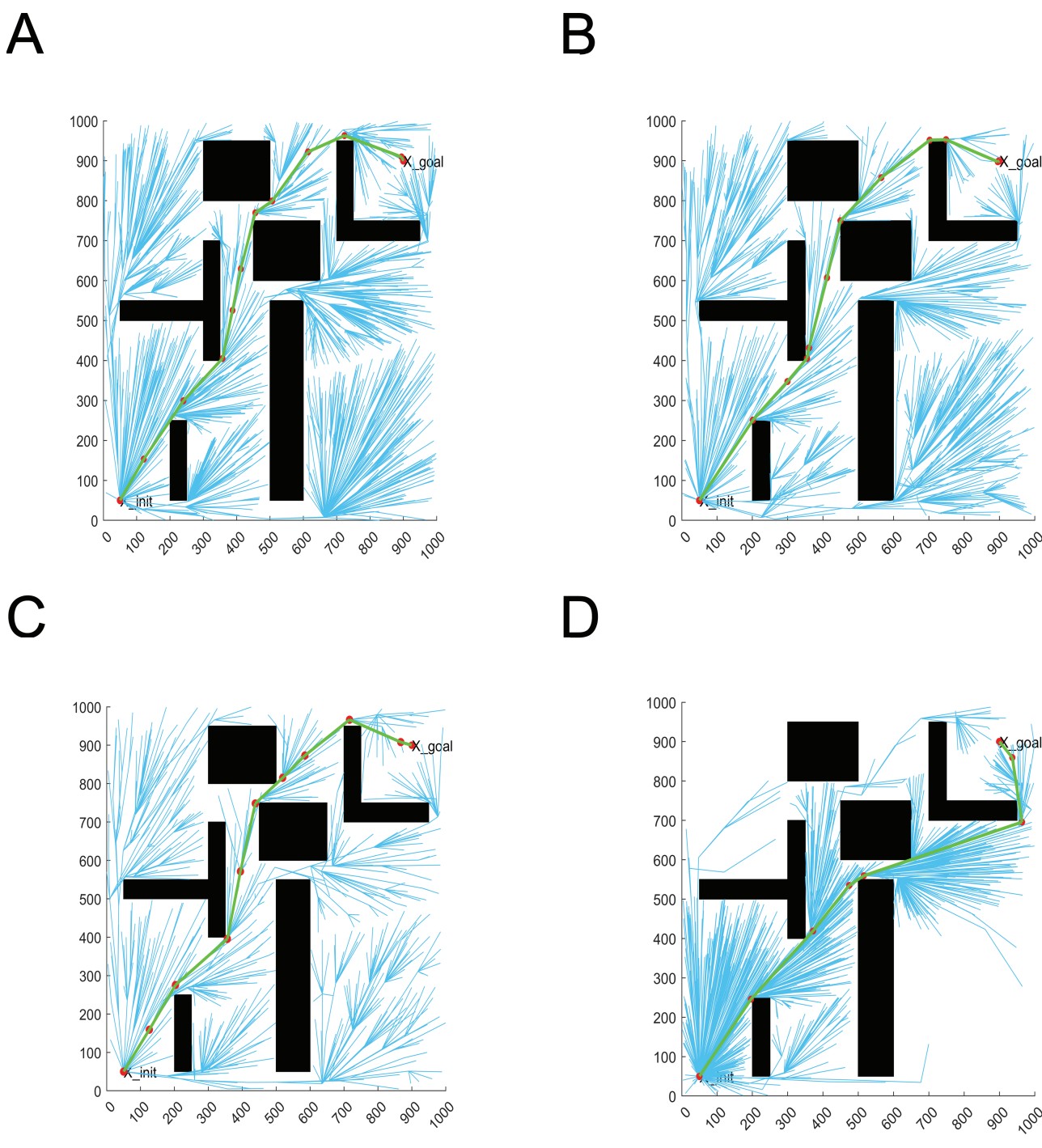

**Fig 16. Planning effect:** (**A**) RRT*. (**B**) MQ-RRT*. (**C**) AS-RRT*. (**D**) LRRT*.

As shown in Fig 20, the relationship between the path cost and the number of iterations, planning time and the path cost of different algorithms are represented in the 3D environment, respectively. From Fig 20A, LRRT* algorithm has faster planning time and smaller

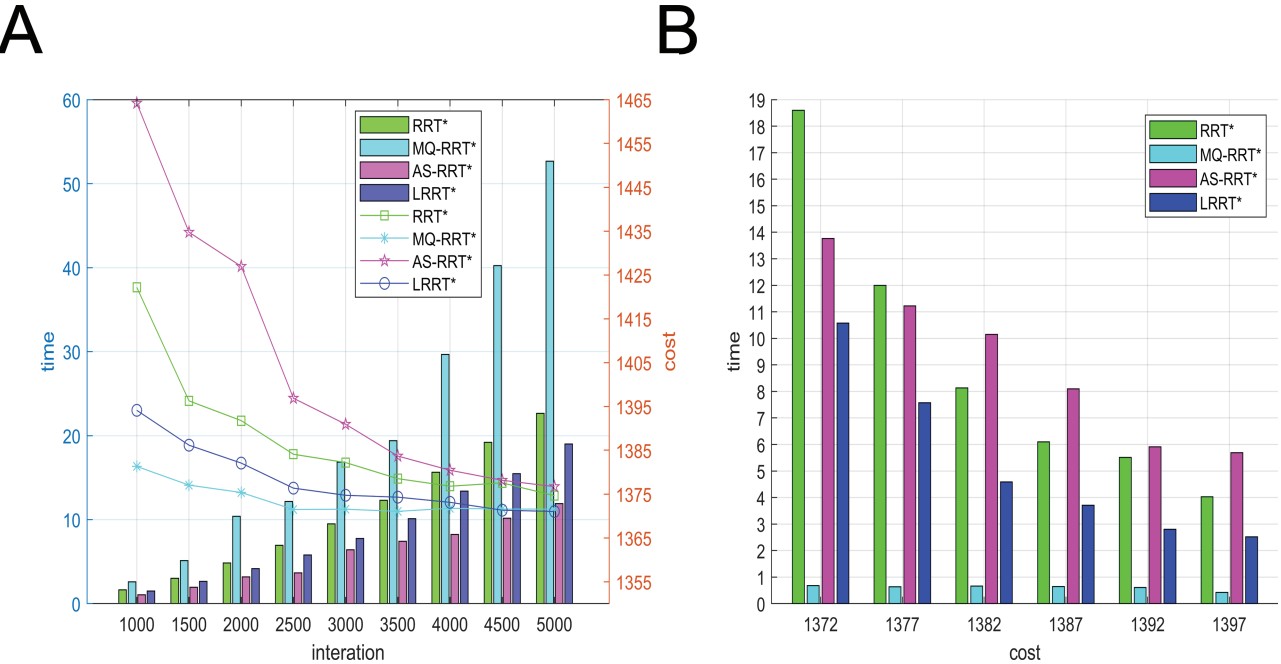

**Fig 17. Comparison of simulation results for complex obstacle environments:** (**A**) Path cost-time-number of iterations. (**B**) Path cost-time.

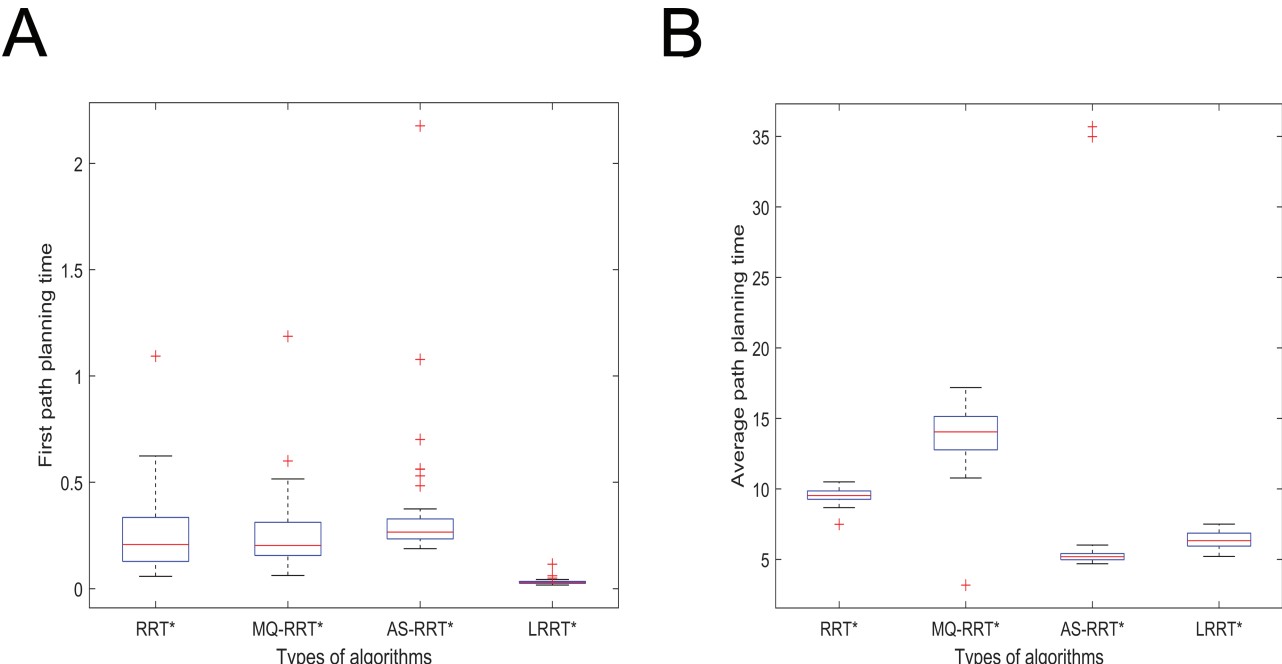

**Fig 18. Planning time for the 3000 iteration:** (**A**) Initial path planning time. (**B**) Average path planning time.

**Table 3. Simulation data of complex obstacle environment.**

| Algorithm | Initial path planning time/s | Average path planning time/s | Number of nodes | Smoothness |
|---|---|---|---|---|
| RRT* | 0.203 | 10.500 | 12.1 | 170.59 |
| MQ-RRT* | 0.281 | 14.842 | 12.6 | 174.72 |
| AS-RRT* | 0.702 | 5.586 | 12.0 | 169.70 |
| LRRT* | 0.115 | 7.276 | 7.5 | 165.07 |
| Improvement/% | 43.3 | 30.7 | 38.0 | 3.4 |

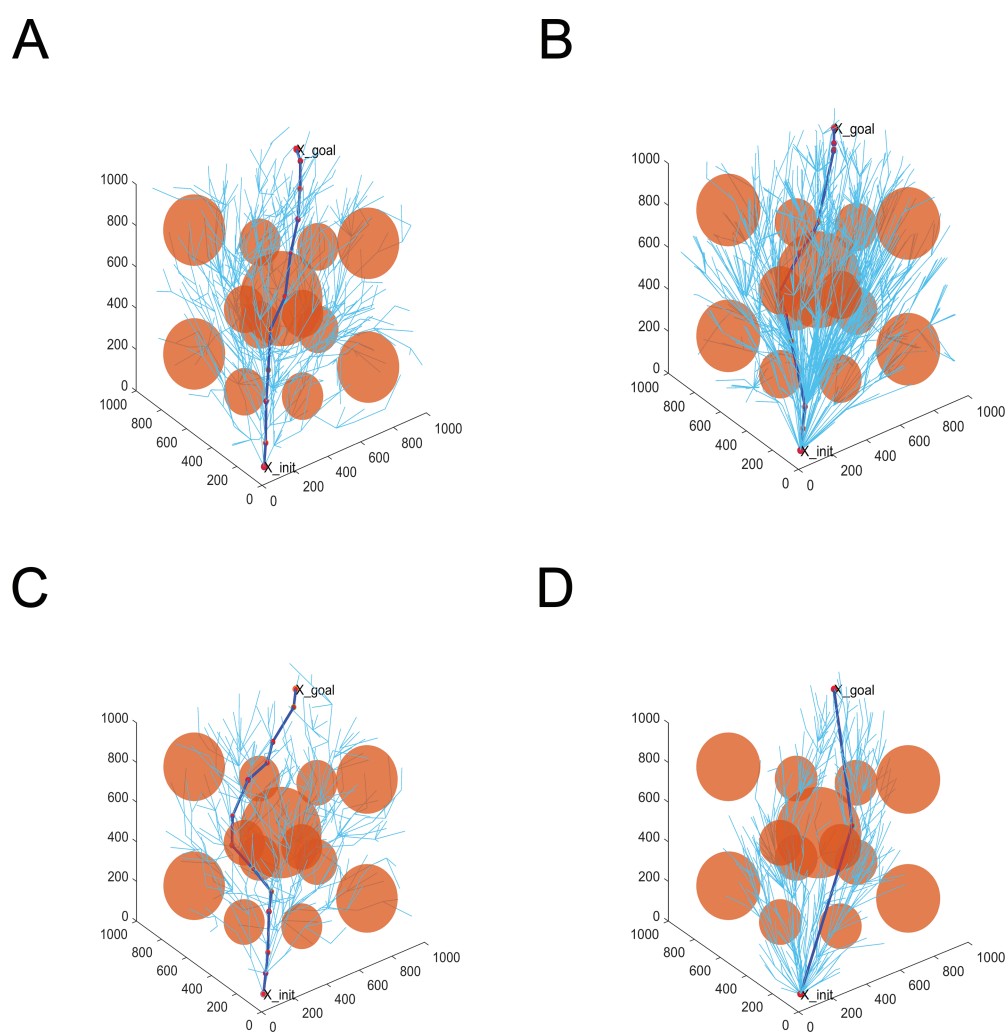

**Fig 19. Planning effect:** (**A**) RRT*. (**B**) MQ-RRT*. (**C**) AS-RRT*. (**D**) LRRT*.

path cost. LRRT* algorithm outperforms RRT* algorithm, MQ-RRT* algorithm and AS-RRT* algorithm not only in planning time but also in path cost. From Fig 20B, LRRT* algorithm takes only a shorter time to plan a path with less path cost. The higher the path cost requirement, the more obvious the contrast is.

The planning time boxplots of the simulation results of different algorithms under 3000 iterations are shown in Fig 21 and the data pairs are shown in Table 4. In Fig 21A, LRRT* algorithm has the lowest box height. It proves that LRRT* algorithm has higher stability in

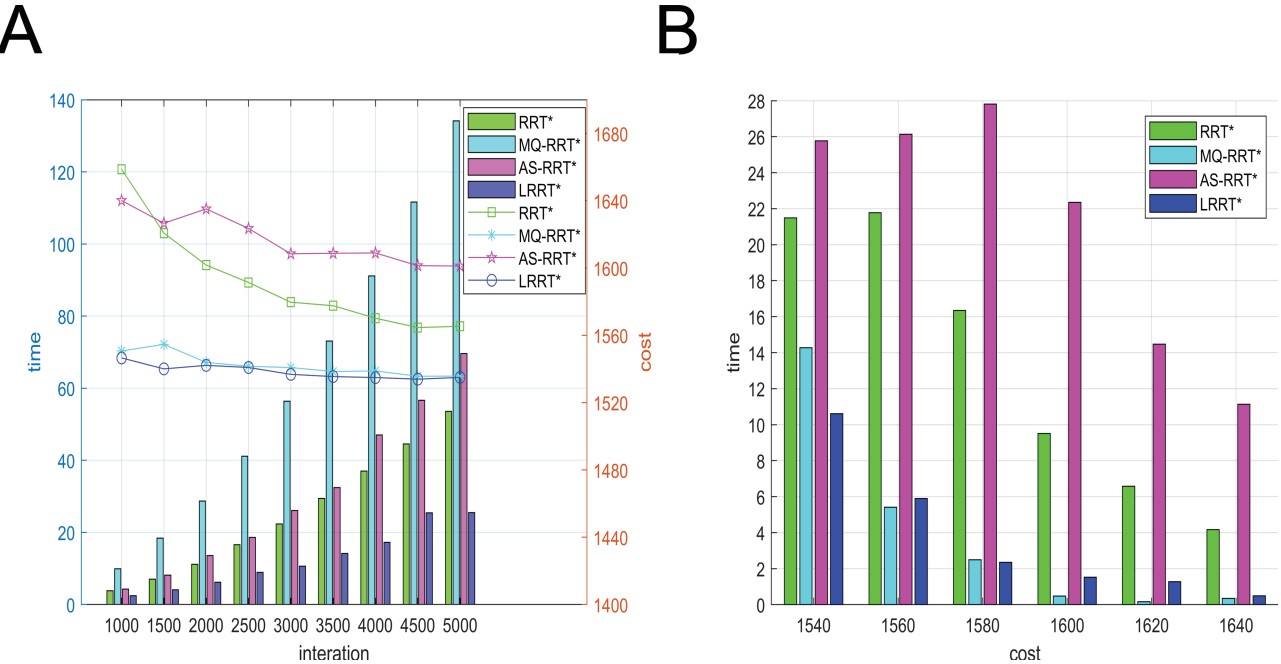

**Fig 20. Comparison of simulation results for 3D environments:** (**A**) Path cost-time-number of iterations. (**B**) Path cost-time.

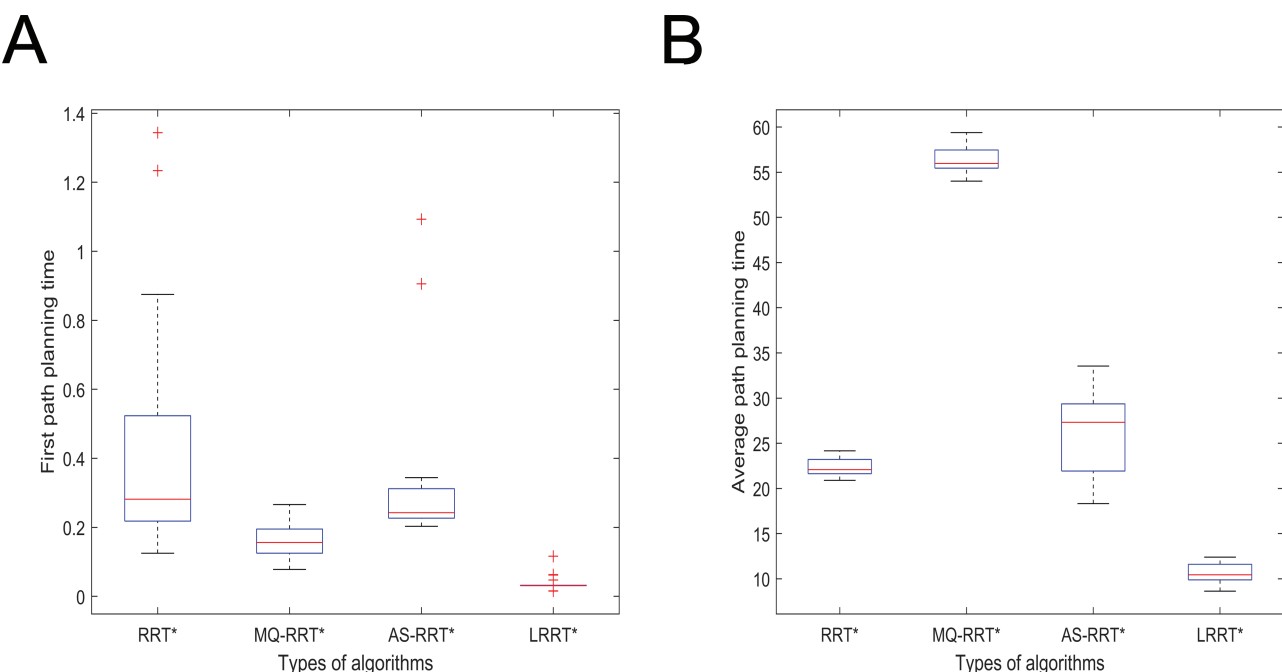

**Fig 21. Planning time for the 3000 iteration:** (**A**) Initial path planning time. (**B**) Average path planning time.

**Table 4. Simulation data of 3D environment.**

| Algorithm | Initial path planning time/s | Average path planning time/s | Number of nodes | Smoothness |
|---|---|---|---|---|
| RRT* | 0.442 | 22.342 | 11.0 | 170.70 |
| MQ-RRT* | 0.162 | 56.365 | 11.6 | 165.24 |
| AS-RRT* | 0.332 | 26.052 | 12.7 | 81.27 |
| LRRT* | 0.036 | 10.648 | 3.8 | 35.14 |
| Improvement/% | 91.9 | 52.3 | 65.4 | 79.4 |

initial path planning time than RRT* algorithm, MQ-RRT* algorithm and AS-RRT* algorithm. It also has the lowest average value. In Fig 21B, the box height of LRRT* algorithm is not much different from RRT* algorithm and MQ-RRT* algorithm. It proves that it also has better stability in average planning time and its average value is also the lowest. In summary, the performance of LRRT* algorithm is also better than RRT* algorithm, MQ-RRT* algorithm and AS-RRT* algorithm in 3-dimensional space. From Table 4, it is obtained that LRRT* algorithm improves 91.9% in initial path planning time, 52.3% in average path planning time, reduces the number of path nodes by 65.4%, and the path smoothness is reduced by 79.4% than RRT* algorithm.

### 4.3 Ablation experiment

Add ablation experiments to verify the impact of the key components of the algorithm on the overall performance, as shown in Fig 22. The main innovations of this thesis are four parts of the LRRT* algorithm with control variables in a two-dimensional complex obstacle environment, as shown in Table 5. The four ablation experiments are respectively represented as LRRT* without target-oriented strategy (Case A), LRRT* without Levy flight strategy (Case B), LRRT* without effective region sampling strategy (Case C), and LRRT* without node rejection strategy (Case D). Since the goal-oriented strategy and the Levy flight strategy are improvements in the fast initial path finding phase, the data for comparison are the initial path planning time and the number of initial path expansion nodes. Combined with Fig 22 and Table 5, the addition of the goal-oriented strategy shortens the initial path planning time. And the addition of the Levy flight strategy increases the effectiveness of nodes. The effective region sampling strategy and node rejection strategy are in the phase of optimizing the initial path, so the data for comparison is mainly the average planning time. From Table 5, it can be seen that adding dynamic region sampling strategy and node rejection strategy shortens the average planning time.

Therefore, RRT* algorithm can be improved comprehensively with the addition of target-oriented strategy, Levy flight strategy, effective region sampling strategy and node rejection strategy. Moreover, after the above simulation verification of LRRT* algorithm in 2D and 3D environments. It shows that LRRT* algorithm is better than RRT* algorithm, MQ-RRT* algorithm and AS-RRT* algorithm in terms of comprehensive performance, and that LRRT* algorithm has a better high-dimensional adaptability.

### 4.4 Path planning simulation of robot arm

The robotic arm model and the collision detection model have been introduced in Section 3. The UR5 robotic arm is modeled in Matlab and the environment such as obstacles is built as shown in Fig 23(A). Two rectangular obstacles and one spherical obstacle are set up in the environment. It is used to verify the ability of the robotic arm to overcome obstacles. The initial joint angle of the manipulator is (0, 0, 0, 0, 0, 0, 0) and the target joint angle

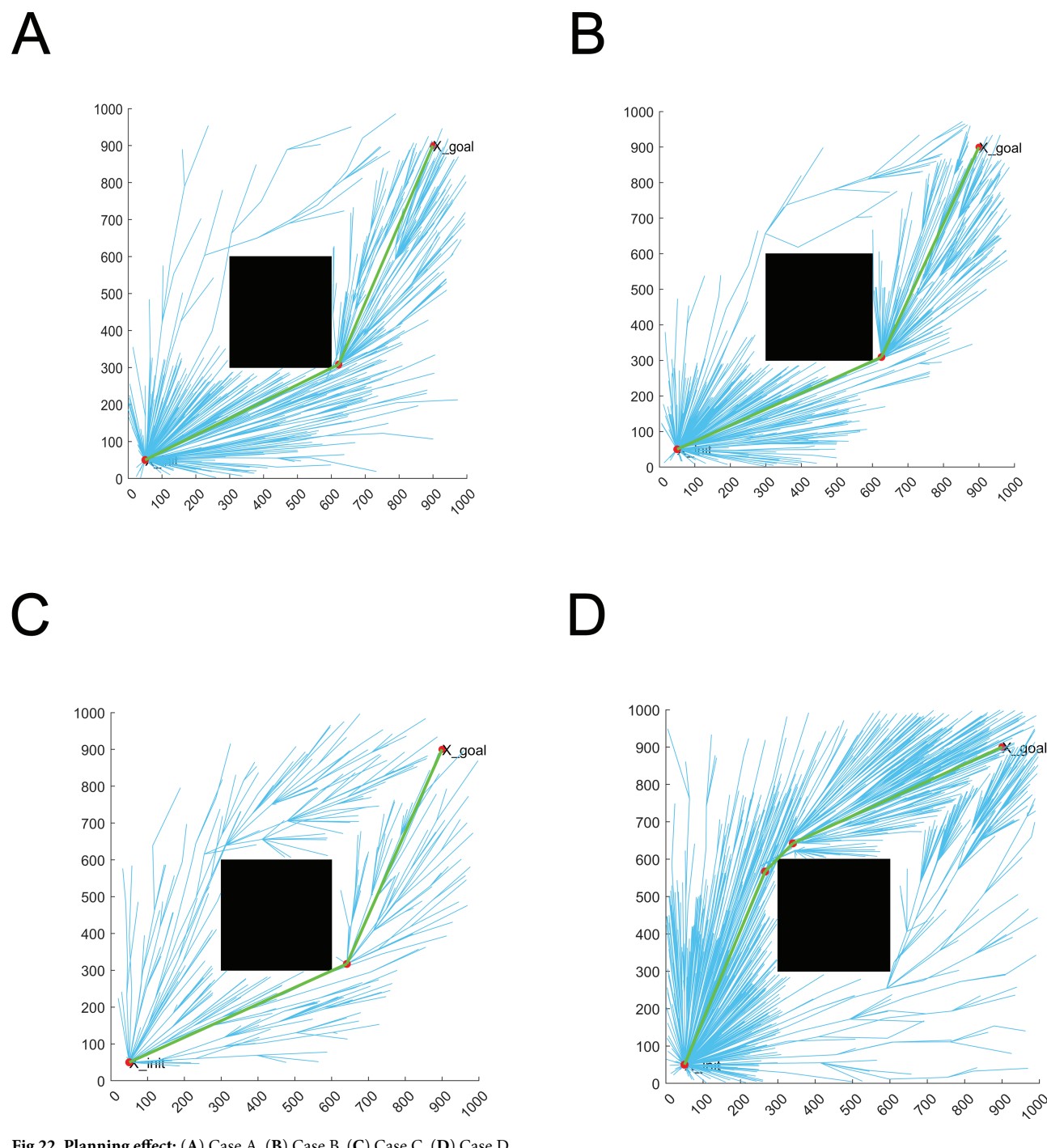

**Fig 22. Planning effect:** (**A**) Case A. (**B**) Case B. (**C**) Case C. (**D**) Case D.

is $\left(\frac{4}{5}\pi, \frac{\pi}{3}, -\frac{2}{3}\pi, 0, 0, 0\right)$. LRRT* algorithm is used for path planning of the robotic arm, as shown in Figure 23. The red broken line represents the planned path, and the green line is the path smoothed by the cubic B-spline curve. It can be seen that LRRT* algorithm has planned a collision-free path. As shown in Fig 23(C), the robotic arm does not

**Table 5. Ablation experiment.**

| Algorithm | Initial path planning time/s | Initial number of path extension nodes | Average path planning time/s |
|---|---|---|---|
| RRT* | 0.244 | 121.7 | 4.766 |
| Case A | 0.239 | 116.7 | 4.501 |
| Case B | 0.253 | 104.4 | 3.450 |
| Case C | 0.203 | 80.1 | 4.368 |
| Case D | 0.196 | 87.2 | 5.711 |
| LRRT* | 0.201 | 81.6 | 4.179 |

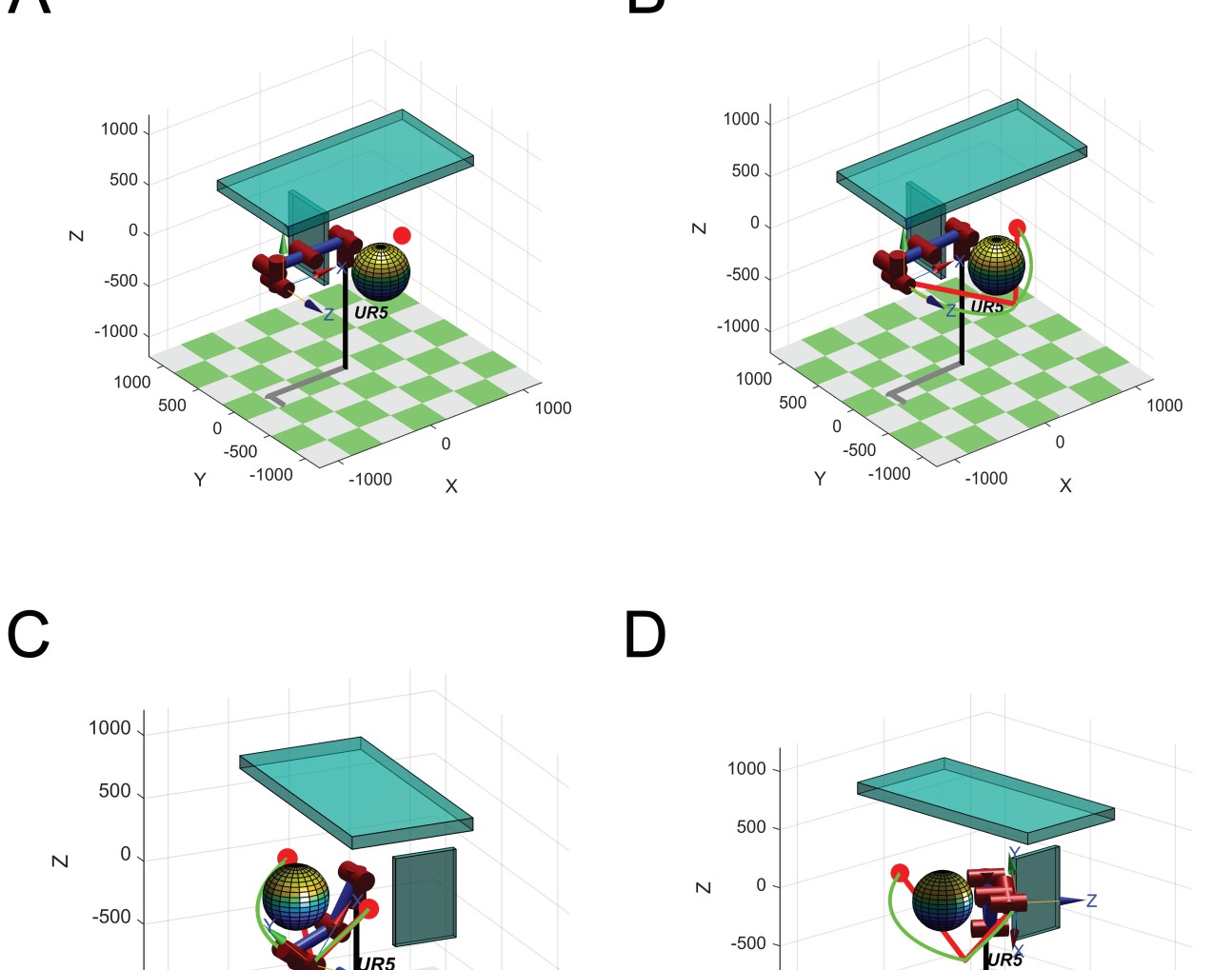

**Fig 23. Operating pose of the robot arm:** (**A**) Robotic arm simulation environment. (**B**) Initial pose. (**C**) Path point pose. (**D**) Target pose.

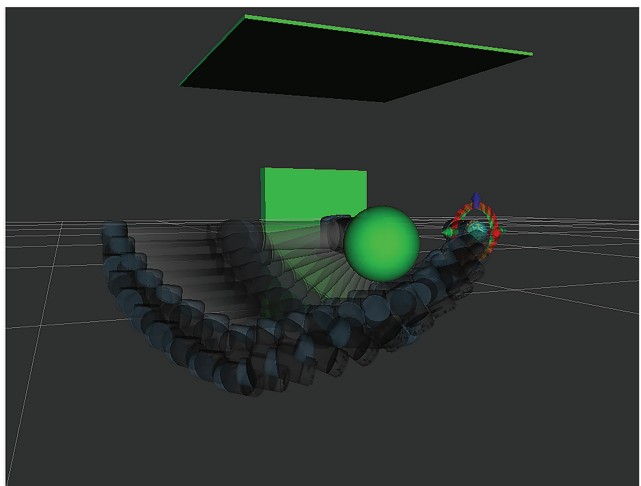

**Fig 24. MoveIt**.

collide with obstacles during operation, which illustrates the effectiveness of the LRRT* algorithm.

To rigorously validate the algorithm's effectiveness, we implemented it within the Robot Operating System (ROS) framework, deployed on an Ubuntu 18.04 virtual machine via VMware Workstation. Collision-free trajectory verification was conducted through simulations using MoveIt!, ROS's motion planning toolkit. As illustrated in Fig 24, the robotic arm's motion trajectory is visualized in the simulation environment. A critical design feature of this framework is its real-time collision detection mechanism: the arm turns red upon intersecting with obstacles. The persistent absence of color transitions to red in Fig 24 conclusively demonstrates that no part of the robotic arm collided with obstacles during the entire motion sequence.

## 5 Conclusion

In this work, an improved algorithm LRRT* algorithm based on RRT* algorithm is proposed for the robotic arm path planning problem. Two-stage planning is used on the basis of RRT* algorithm: fast finding initial path and optimizing initial path. In the stage of quickly finding the initial path, the goal-oriented strategy and Levy flight strategy are introduced. They can quickly find a collision-free initial path. In the phase of optimizing the initial path, effective region sampling is introduced to continuously optimize the initial path and quickly converge to the optimal path. Node rejection strategy is introduced to reduce the number of collision detection times during pruning and shorten the time. Finally, the greedy idea is added to reduce the number of path points, and 3 times B-spline curve is added to smooth the path. Simulations are performed in both 2D and 3D environments. The simulation results show that in the 2D environment, the proposed LRRT* algorithm reduces the initial path planning time by 17.6%, the number of initial path expansion nodes by 32.9%, and the average planning time by 12.3%. In the 3D environment, the proposed LRRT* algorithm reduces the initial path planning time by 91.9%, the number of initial path expansion nodes by 52.3%, and the average planning time by 65.5%. Finally, it is verified in Matlab that the LRRT* algorithm for UR5 robotic arm can plan a collision-free path. The LRRT* algorithm is effective.

Improved LRRT* algorithm in this work is mainly used for the picking robotic arm. It can also be used for cart path planning and UAV path planning. The next step is to transplant the algorithm to the ROS operating system, conduct simulation debugging of the real robot arm, and further optimize the algorithm.

## Author contributions

**Conceptualization:** Yu Gu.

**Data curation:** Hua Luo.

**Formal analysis:** Wenbin Gong.

**Investigation:** Yutao Jiang.

**Methodology:** Yu Gu.

**Project administration:** Tangju Yuan.

**Resources:** Longzhou Cao.

**Software:** Yu Gu.

**Supervision:** Hongbing Li.

**Validation:** Wei Zhang.

**Writing – original draft:** Yu Gu.

**Writing – review & editing:** Hongbing Li.

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
