## [Decision Letter · Decision Letter 0]

PONE-D-25-04031LRRT*: A robotic arm path planning algorithm based on an improved Levy Flight Strategy with effective region sampling RRT*PLOS ONE

Dear Dr. Gu,

Thank you for submitting your manuscript to PLOS ONE. After careful consideration, we feel that it has merit but does not fully meet PLOS ONE’s publication criteria as it currently stands. Therefore, we invite you to submit a revised version of the manuscript that addresses the points raised during the review process.

We look forward to receiving your revised manuscript.

Kind regards,

Jiaqi Miao

Guest Editor

PLOS ONE

Journal Requirements:

The work was supported by National Key R&D Program of Chin(2021YFB3901400)�Natural Science Foundation of Chongqing municipality (2022NSCQ-MSX4084), Scientific and Technological Research Program of Chongqing Municipal Education Commission (KJZD-M202201204, KJZD-M202301203, KJQN202301215, KJQN202401226, 22SKGH333), Open Fund of Chongqing Key Laboratory of Geo-environment Monitoring and Disaster Early Warning of Three Gorges Reservoir Area (MP2020B0202), Science and Technology Innovation Smart Agriculture Project of Wanzhou District Science and Technology Bureau (2022-17), Cultivation project of Chongqing social science planning project (2019PY52).

6. We note that your Data Availability Statement is currently as follows: All relevant data are within the manuscript and its Supporting Information files.

Reviewers' comments:

Reviewer's Responses to Questions

**Comments to the Author**

1. Is the manuscript technically sound, and do the data support the conclusions?

Reviewer #1: Partly

Reviewer #2: Yes

Reviewer #3: Yes

2. Has the statistical analysis been performed appropriately and rigorously? 

Reviewer #1: Yes

Reviewer #2: Yes

Reviewer #3: Yes

3. Have the authors made all data underlying the findings in their manuscript fully available?

Reviewer #1: Yes

Reviewer #2: Yes

Reviewer #3: Yes

4. Is the manuscript presented in an intelligible fashion and written in standard English?

Reviewer #1: Yes

Reviewer #2: Yes

Reviewer #3: Yes

5. Review Comments to the Author

Reviewer #1: The paper proposes an improved LRRT* algorithm to address the limitations of traditional RRT* algorithms, such as low node efficiency and slow convergence, in robotic arm path planning. By integrating a Levy flight strategy and effective region sampling, the authors aim to enhance path planning efficiency and quality in complex environments. While the research motivation is clear and the experimental data are comprehensive, several critical issues remain unresolved:

1. The main innovation of the paper lies in the introduction of the Levy flight strategy within the RRT* algorithm. However, the paper provides limited introduction to this strategy and rarely involves experiences from other similar path planning scenarios where related strategies have been used to optimize algorithms.

2. During the introduction of the Levy flight strategy, the parameter selection phase (β=1.5) is based solely on empirical values, without mentioning the reference sources for these values, nor providing other comparative experiments to explain the rationale behind this parameter selection.

3. The paper seems to emphasize planning efficiency and the time required for planning in its evaluation of the algorithm. However, since the algorithm focuses on the end-effector trajectory planning of the robotic arm, the executability and smoothness of the path in this scenario might be more critical compared to path planning for UAVs. The paper could provide more dimensions of evaluation criteria to further assess the rationality of the algorithm optimization results.

4. The paper's collision detection is limited to modeling and model simplification. Is there other data to support that collision detection was performed on any part of the robotic arm during the experiments, or are there other evaluation indicators in the experimental results that can demonstrate the algorithm selected path planning results that ensure no collisions occur on any part of the robotic arm?

Reviewer #2: It is an excellent piece of work. Congratulations on the quality of your research and the effort you’ve put into it. As one of the reviewers, I sincerely appreciate the depth of analysis, the clarity of presentation, and the contribution this paper makes to the field. I encourage you to continue refining and developing your ideas, as they have great potential for impact within the academic community.

Reviewer #3: The contribution of this manuscript is clear. Therefore, my recommendation is to accept (Minor Revision) this manuscript that has Ref. No.: PONE-D-25-04031 after several adjustments must be made before publication.

My specific comments are:

1- In the abstract, the result of this work must be described briefly with data in order to show the effectiveness of the proposed work.

2- The author did not describe the drawbacks of each conventional technique in the introduction paragraph.

3- Please include the references for all equations.

4- In Tables 2, 3, and 4, why the number of nodes is not integer number?

5- In Tables 2, 3, and 4, please add another column to show an enhancement percentage (%).

6. PLOS authors have the option to publish the peer review history of their article (what does this mean?). If published, this will include your full peer review and any attached files.

Reviewer #1: No

Reviewer #2: No

Reviewer #3: No

---

## [Author Response · Author response to Decision Letter 1]

8 Apr 2025

Dear Editor,

Please find our revised manuscript entitled "LRRT*: A robotic arm path planning algorithm based on an improved Levy Flight Strategy with effective region sampling RRT* " (Original Manuscript ID: PONE-D-25-04031)

We would like to take this opportunity to express our sincere gratitude to the editors, as well as the profound opinions and constructive suggestions of the reviewers. We have carefully considered all the comments from the reviewers. In the revised version, significant changes have been made. Below, we provide our response to these comments. The italicized sentences are comments, while the rest are our responses.

We noticed that you mentioned in your email that the funding information provided in the "Funding Information" and "Financial Disclosure" sections does not match, and we hereby restate the funding information: The work was supported by National Key R&D Program of China (2021YFB3901400)�Natural Science Foundation of Chongqing municipality (2022NSCQ-MSX4084), Scientific and Technological Research Program of Chongqing Municipal Education Commission (KJZD-M202201204, KJZD-M202301203, KJQN202301247, KJQN202401212, KJQN202301215�KJQN202401226, 22SKGH333), Open Fund of Chongqing Key Laboratory of Geo-environment Monitoring and Disaster Early Warning of Three Gorges Reservoir Area (MP2020B0202). The funders had no role in study design, data collection and analysis, decision to publish, or preparation of the manuscript. In addition, I have included in my manuscript all the original data needed to reproduce my findings.

We believe these changes have effectively improved the quality of the paper. Thank you and the reviewers for your time and efforts.

Sincerely

Yu Gu1, Hua Luo2*, Wenbin Gong1, Yutao Jiang1, Tangju Yuan1, Longzhou Cao1*, Hongbing Li2,3, Wei Zhang1

1.Chongqing Key Laboratory of Geological Environmental Monitoring and Disaster Early Warning in the Three Gorges Reservoir Area, Chongqing Three Gorges University, Wanzhou, Chongqing, 404120, China

2.Internet of Things and Intelligent Control Technology Chongqing Engineering Research Center, Chongqing Three Gorges University, Wanzhou, Chongqing, 404120, China

3.Chongqing Municipal Key Laboratory of Intelligent Information Processing and Control, Chongqing Three Gorges University, Wanzhou, Chongqing, 404120, China

* Corresponding author. Email: 1915854548@qq.com; 29209645@qq.com

Reviewer#1, Concern # 1 : The main innovation of the paper lies in the introduction of the Levy flight strategy within the RRT* algorithm. However, the paper provides limited introduction to this strategy and rarely involves experiences from other similar path planning scenarios where related strategies have been used to optimize algorithms.

Authors response: Thank you for reviewing our manuscript and providing valuable suggestions. We have carefully considered your feedback and added a theoretical introduction to the Levy flight strategy in Section 3.3.2. Additionally, we have incorporated two Levy flight strategies for the research and analysis of path planning.

Authors action: The Levy flight is a specialized random walk model used to characterize movement patterns with long-tailed distributions. In Levy flights, individuals perform stochastic movements in space, where both step lengths and directions are governed by the Levy distribution—a probability distribution exhibiting heavy-tailed characteristics. Its probability density function follows a power-law relationship, indicating that Levy-distributed motion exhibits significantly higher frequencies of large-step events (i.e., long-distance displacements) compared to Gaussian or other conventional distributions.

In addressing path planning challenges under static and dynamic environments, He Jianchen et al. integrated the Levy flight strategy with the Dung Beetle Optimizer (DBO) and Dynamic Window Approach (DWA). Specifically, they modified the position update formulas for breeding and foraging dung beetles in the DBO algorithm using Levy flight dynamics, thereby enhancing the algorithm's exploration capability and adaptability[28]. Correspondingly, Niu Yanbiao et al. tackled the issue of poor dynamic obstacle avoidance in UAV path planning by proposing an enhanced Sand Cat Swarm Optimization (SCSO) algorithm. This improved framework incorporates an adaptive social neighborhood search mechanism and Levy flight strategies, which collectively elevate solution quality through refined exploration-exploitation balance[29].

[28] Jiachen, H., Li-hui, F. Robot path planning based on improved dung beetle optimizer algorithm. J Braz. Soc. Mech. Sci. Eng; 46, 235, 2024.

[29] Yanbiao, N., Xuefeng, Y., Yongzhen, W., Yanzhao, N. 3D real-time dynamic path planning for UAV based on improved interfered fluid dynamical system and artificial neural network. Advanced Engineering Informatics; 59, 102306, 2024.

Reviewer#1, Concern # 2 : During the introduction of the Levy flight strategy, the parameter selection phase (β=1.5) is based solely on empirical values, without mentioning the reference sources for these values, nor providing other comparative experiments to explain the rationale behind this parameter selection.

Authors response: Thank you for your valuable suggestions and feedback on our paper. We have included the source of beta value in Section 3.3.2, cited several papers in multiple fields, and conducted comparative experiments on the selection of beta in these papers. It is found that when β=1.5, the algorithm achieves the best balance between global search ability and convergence speed. Moreover, when β=1.5 in the field of manipulator, the system has the best performance in trajectory tracking accuracy and vibration suppression. All these prove that it is reasonable, so no comparative experiment is done in this paper.

Authors action: where is the flight path, and β takes the value in the range of [0,2], and is generally taken as β=1.5[31]-[33].

[31] Hüseyin, H., Harun, U. A novel particle swarm optimization algorithm with Levy flight. Applied Soft Computing; 23, 333-345, 2014.

[32] Ahmad, MA., Mohd Rashid, M., Sulaiman, MH., Suid, MH., Mohd Tumari, MZ. Levy Flight Safe Experimentation Dynamics Algorithm for Data-Based PID Tuning of Flexible Joint Robot. 2020 IEEE 10th Symposium on Computer Applications & Industrial Electronics (ISCAIE), Malaysia, 108-112, 2020.

[33] Chawla, M., Duhan, M. Levy Flights in Metaheuristics Optimization Algorithms – A Review. Applied Artificial Intelligence;32, 802–821, 2018.

Reviewer#1, Concern # 3 : The paper seems to emphasize planning efficiency and the time required for planning in its evaluation of the algorithm. However, since the algorithm focuses on the end-effector trajectory planning of the robotic arm, the executability and smoothness of the path in this scenario might be more critical compared to path planning for UAVs. The paper could provide more dimensions of evaluation criteria to further assess the rationality of the algorithm optimization results.

Authors response: Thank you for your valuable suggestions on the algorithm evaluation dimension. We fully agree that the path planning of robotic arms needs to pay attention to engineering indicators such as executability and smoothness. We have added the path smoothness evaluation index in the experimental chapter, and the path execution can be seen by the collision-free operation of the robot arm.

Authors action: The smoothness of a path can be quantified by measuring the angular variation between consecutive path points. To systematically evaluate this property, we propose a path smoothness metric , defined as:

Where denotes the angle between the (i−1)-th and i-th path segments. A higher value indicates increased path tortuosity, while lower values correspond to smoother trajectories.

Reviewer#1, Concern # 4 : The paper's collision detection is limited to modeling and model simplification. Is there other data to support that collision detection was performed on any part of the robotic arm during the experiments, or are there other evaluation indicators in the experimental results that can demonstrate the algorithm selected path planning results that ensure no collisions occur on any part of the robotic arm?

Authors response: Thank you for your constructive feedback on our manuscript. To rigorously validate the collision-free operation of the robotic arm throughout its entire motion sequence, we have integrated the proposed algorithm into the Robot Operating System (ROS) under a Linux environment. The trajectory of the manipulator can be seen in the Moveit simulation platform.

Authors action: To rigorously validate the algorithm's effectiveness, we implemented it within the Robot Operating System (ROS) framework, deployed on an Ubuntu 18.04 virtual machine via VMware Workstation. Collision-free trajectory verification was conducted through simulations using MoveIt!, ROS's motion planning toolkit. As illustrated in Figure 24, the robotic arm's motion trajectory is visualized in the simulation environment. A critical design feature of this framework is its real-time collision detection mechanism: the arm turns red upon intersecting with obstacles. The persistent absence of color transitions to red in Figure 24 conclusively demonstrates that no part of the robotic arm collided with obstacles during the entire motion sequence.

Reviewer#2 : It is an excellent piece of work. Congratulations on the quality of your research and the effort you’ve put into it. As one of the reviewers, I sincerely appreciate the depth of analysis, the clarity of presentation, and the contribution this paper makes to the field. I encourage you to continue refining and developing your ideas, as they have great potential for impact within the academic community.

Authors response: Dear reviewer, thank you so much for your reviewing! We deeply appreciate your recognition of our research work.

Authors action: The manuscript does not need to be updated.

Reviewer#3, Concern # 1 : In the abstract, the result of this work must be described briefly with data in order to show the effectiveness of the proposed work.

Authors response: Thank you for reviewing our manuscript and for your valuable suggestions. We have taken your comments into account and, in addition to the existing path length and time optimization data, have added path smoothness data to more intuitively demonstrate the effectiveness of the proposed method.

Authors action: We have augmented the Abstract:

In 2D and 3D environments, the LRRT* algorithm reduces the initial path planning time by 17.6% and 91.9% respectively compared to the RRT* algorithm, and shortens the average planning time by 12.3% and 65.5%, and the path smoothness is 3.4% and 79.4% shorter respectively.

Reviewer#3, Concern # 2 : The author did not describe the drawbacks of each conventional technique in the introduction paragraph.

Authors response: Thank you for your valuable suggestions and feedback on our paper. We have analyzed in detail the limitations of traditional methods in the path planning of robotic arms in the introduction.

Authors action: Swarm intelligence-based optimization algorithms exhibit notorious susceptibility to local optima entrapment in complex obstacle environments, compounded by excessive parameter sensitivity. For instance, in robotic manipulators with high degrees of freedom (DoFs), the dimensional explosion of solution space fundamentally undermines population-based methods like Genetic Algorithms (GAs). These algorithms necessitate maintaining large-scale populations (typically hundreds to thousands of individuals), with each iteration requiring selection, crossover, and mutation operations—a process whose temporal complexity escalates exponentially with dimensionality, thereby failing to meet real-time constraints.

The Artificial Potential Field (APF) approach faces dual limitations: motion stagnation occurs when the resultant vector of attractive and repulsive fields nullifies at non-target configurations, while its hyper-sensitive parameter tuning requirements and poor generalizability hinder practical deployment. Furthermore, graph-search algorithms like A* and Dijkstra encounter dimensional catastrophe in high-dimensional configuration spaces. The discretized node count grows exponentially with dimensionality, rendering these methods computationally prohibitive for robotic applications requiring.

Reviewer#3, Concern # 3 : Please include the references for all equations.

Authors response: We have methodically implemented citation protocols for all non-original formulations, ensuring full traceability of theoretical foundations. Specifically:Goal-oriented formulations in Section 3.3.1 are derived from reference [27]. Lévy flight-driven stochastic operators in Section 3.3.2 follow the mathematical framework established in [30]. Bézier curve parameterizations for trajectory smoothing in Section 3.3.6 adhere to the geometric modeling principles documented in [34].

Authors action:

Therefore, a goal-oriented strategy is introduced, as shown in Eq(4)[27].

To calculate the search path for Levy flight, the formula proposed by Mantegna for modeling the Levy flight path is usually used, as shown in Eq(5)[30].

N times B-spline curve equation is shown in Eq(10)[34].

[27] Fu, L., Lin, X., Lou, Y. Adaptive Goal-Biased Bi-RRT for Online Path Planning of Robotic Manipulators. Intelligent Robotics and Applications; 14267,2023.

[30] Jiachen, H., Li-hui, F. Robot path planning based on improved dung beetle optimizer algorithm. J Braz. Soc. Mech. Sci. Eng; 46, 235, 2024.

[34] kheireddine, C., Yassine, A., Fawzi, S. A robust synergetic controller for Quadrotor obstacle avoidance using Bézier curve versus B-spline trajectory generation. Intel Serv Robotics 15, 143–152, 2022.

Reviewer#3, Concern # 4 : In Tables 2, 3, and 4, why the number of nodes is not integer number?

Authors response: Thank you for your comments. We explain the reasons for this result as follows: The non-integer number of nodes is derived from the statistical average of 50 experiments, and one decimal place is reserved to reflect the statistical fluctuation, which is explained at the time of experiment setup.

Authors action: The manuscript does not need to be updated.

Reviewer#3, Concern # 5 : In Tables 2, 3, and 4, please add another column to show an enhancement percentage (%).

Authors response: Thank you for your valuable suggestions and feedback on our paper. We have updated Tables 2, 3, and 4 with the addition of a percentage improvement (%) column to visually show the performance improvement of the LRRT* algorithm over the benchmark method.

Authors action: We updated the manuscript by “Table 2, 3, and 4”.

---

## [Decision Letter · Decision Letter 1]

LRRT*: A robotic arm path planning algorithm based on an improved Levy Flight Strategy with effective region sampling RRT*

PONE-D-25-04031R1

Dear Dr. Gu,

We’re pleased to inform you that your manuscript has been judged scientifically suitable for publication and will be formally accepted for publication once it meets all outstanding technical requirements.

Kind regards,

Jiaqi Miao

Guest Editor

PLOS ONE

Additional Editor Comments (optional):

Reviewers' comments:

Reviewer's Responses to Questions

**Comments to the Author**

1. If the authors have adequately addressed your comments raised in a previous round of review and you feel that this manuscript is now acceptable for publication, you may indicate that here to bypass the “Comments to the Author” section, enter your conflict of interest statement in the “Confidential to Editor” section, and submit your "Accept" recommendation.

Reviewer #1: All comments have been addressed

Reviewer #3: All comments have been addressed

2. Is the manuscript technically sound, and do the data support the conclusions?

Reviewer #1: Yes

Reviewer #3: Yes

3. Has the statistical analysis been performed appropriately and rigorously? 

Reviewer #1: Yes

Reviewer #3: Yes

4. Have the authors made all data underlying the findings in their manuscript fully available?

Reviewer #1: Yes

Reviewer #3: Yes

5. Is the manuscript presented in an intelligible fashion and written in standard English?

Reviewer #1: Yes

Reviewer #3: Yes

6. Review Comments to the Author

Reviewer #1: (No Response)

Reviewer #3: The contribution of this revised manuscript is clear and good. All comments from the reviewers were very good, and the author’s answer was very good and satisfied. Therefore, my recommendation is to accept the revised manuscript that has Ref. No.: PONE-D-25-04031R1 for publication.

7. PLOS authors have the option to publish the peer review history of their article (what does this mean?). If published, this will include your full peer review and any attached files.

Reviewer #1: No

Reviewer #3: No

---

## [Editor Report · Acceptance letter]

PONE-D-25-04031R1

PLOS ONE

Dear Dr. Gu,

I'm pleased to inform you that your manuscript has been deemed suitable for publication in PLOS ONE. Congratulations! Your manuscript is now being handed over to our production team.

Kind regards,

on behalf of

Mr Jiaqi Miao

Guest Editor

PLOS ONE